# Photoinduced loading of electron-rich Cu single atoms by moderate coordination for hydrogen evolution

Weiwei Fu [1,5], Jin Wan[1,5], Huijuan Zhang[1,5], Jian Li[2], Weigen Chen[2], Yuke Li [3], Zaiping Guo [4] & Yu Wang [1,2] ✉

Single-atom catalysts offer maximal atom utilization efficiencies and high-electronegativity heteroatoms play a crucial role in coordinating reactive single metal atoms to prevent agglomeration. However, these strong coordination bonds withdraw electron density for coordinated metal atoms and consequently affect their catalytic activity. Herein we reveal the high loading (11.3 wt%) and stabilization of moderately coordinated Cu-P₃ structure on black phosphorus support by a photochemical strategy with auxiliary hydrogen. Single-atom Cu sites with an exceptional electron-rich feature show the $\triangle G_{H^*}$ close to zero to favor catalysis. Neighboring Cu atoms work in synergy to lower the energy of key water adsorption and dissociation intermediates. The reported catalyst shows a low overpotential of only 41 mV at 10 mA cm$^{-2}$ and Tafel slope of 53.4 mV dec$^{-1}$ for the alkaline hydrogen evolution reaction, surpassing both isolated Cu single atoms and Cu nanoclusters. The promising materials design strategy sheds light on the design and fabrication of high-loading single metal atoms and the role of neighboring single atoms for enhanced reaction kinetics.

Single-atom catalysts (SACs) with 100% metal dispersity offer the maximum atom efficiency to create cost-effective catalysts[1–5]. The SACs possess distinct active sites or catalytic pathways different from those of conventional metal catalysts, exhibiting superior activity and selectivity towards oxygen reduction[6–8], CO oxidation[1,9], hydrogenation reactions[5,10], and other important reactions[11–16]. During the past decade, various concepts for atomic dispersion of metals on solid supports have emerged, such as the utilization of vacancy defects on supports[17–19], fabrication of metal-organic frameworks (MOFs)[20,21], spatial confinement in zeolites[22,23], and enhancement of the metal-support interactions[3,24–26]. However, these routes involve fussy synthetic steps and sensitive conditions, including adsorption and reduction of metal precursors[27,28]. Moreover, especially under high

amounts of metal precursors or high-temperature pyrolysis, these methods still cannot strictly exclude metal aggregation, resulting in low reproducibility[29–32]. Currently, most SACs have reported a low metal loading, so the development of a practical and direct approach for constructing SACs with high metal loading is particularly attractive in the field[33].

The strong coordination bonds between metal species and coordinating atoms with lone pairs of electrons such as N, O, and S play a crucial role in preventing atomic agglomeration to achieve atomic-level dispersed catalytic structures[12,34–37]. The formation of heteroatom bonds changes the electronic structure (*d*-band center) of metal atoms by ligand effects. For some catalytic reactions, the electron state of catalytic centers is highly related to the binding energy of

[1]The School of Chemistry and Chemical Engineering, State Key Laboratory of Power Transmission Equipment & System Security and New Technology, Chongqing University, 174 Shazheng Street, Shapingba District, Chongqing City 400044, PR China. [2]The school of Electrical Engineering, Chongqing University, 174 Shazheng Street, Shapingba District, Chongqing City 400044, China. [3]Department of Chemistry, Centre for Scientific Modeling and Computation, Chinese University of Hong Kong, Shatin 999077, Hong Kong. [4]School of Chemical Engineering and Advanced Materials, University of Adelaide, Adelaide 5005, Australia. [5]These authors contributed equally: Weiwei Fu, Jin Wan, Huijuan Zhang. ✉e-mail: wangy@cqu.edu.cn

adsorbate[38,39]. Especially for catalytic reduction reactions, the electron-rich centers facilitate the reduction of reactant[40–42]. However, the high-electronegative atoms probably cause the coordinated metal atoms to form a highly oxidized state, resulting in poisoning or deactivating the single-atomic active centers, while too weak interaction between them makes it difficult to stabilize single atoms. Therefore, the construction of support materials with a well-defined structure to stabilize the catalytic metal atoms in the absence of strong heteroatom coordination is a critical challenge.

Herein, we report a room-temperature photochemical strategy with hydrogen auxiliary to produce stable and high-loading SACs (e.g., Cu, Co) with non-strongly coordinated M-P$_3$ structure on two-dimensional (2D) black phosphorus (BP) support without the aid of heteroatom. Visible light-induced formation of hydrogen radicals (H) on the BP layers is shown to be critical for preparing high-loading neighboring Cu single atoms (n-Cu/BP) with metal-atomic loading up to 11.3%. The electron-rich metal-atomic centers with optimized electronic properties and chemical activity were constructed by the low-electronegativity P atoms. Theoretical calculations have revealed that the active sites of single-atom Cu show the $\triangle G_{H^*}$ close to zero for hydrogen evolution. A neighboring Cu SAC exhibits high water

dissociation activity, significantly surpassing both isolated Cu SAC and Cu nanoclusters. When the Co atom is introduced, n-Cu/BP still shows a greater turnover frequency (TOF) of 0.53 H$_2$ s$^{-1}$ than that of neighboring Co SAC (n-Cu/BP, 0.12 H$_2$ s$^{-1}$) and bi-atomic CuCo/BP (0.17 H$_2$ s$^{-1}$). This work provides a simple design and fabrication of high-loading SACs and the essentiality of neighboring single atom configurations for enhanced kinetics.

## Results

### Synthesis and characterization of catalysts

The synthesis of Cu/BP and Co/BP catalysts is schematically shown in Fig. 1a and Supplementary Fig. 1. BP layers were synthesized by liquid exfoliation of bulk BP in the N-methyl-2-pyrrolidone (NMP) under an Ar atmosphere. The transmission electron microscopy (TEM) image and the redshifted Raman signals of BP in Fig. 1b and Supplementary Fig. 2 conform to the successful exfoliation of bulk BP. The exfoliated BP layers show a distinct 2D morphology with an average thickness of ~2.4 nm by atomic force microscopy (AFM), corresponding to four layers (Supplementary Fig. 3)[43,44]. In the BP structure, every phosphorus atom exposes a pair of lone pair electrons. The zeta-potential of BP nanosheets dispersed in NMP shows a negative surface potential

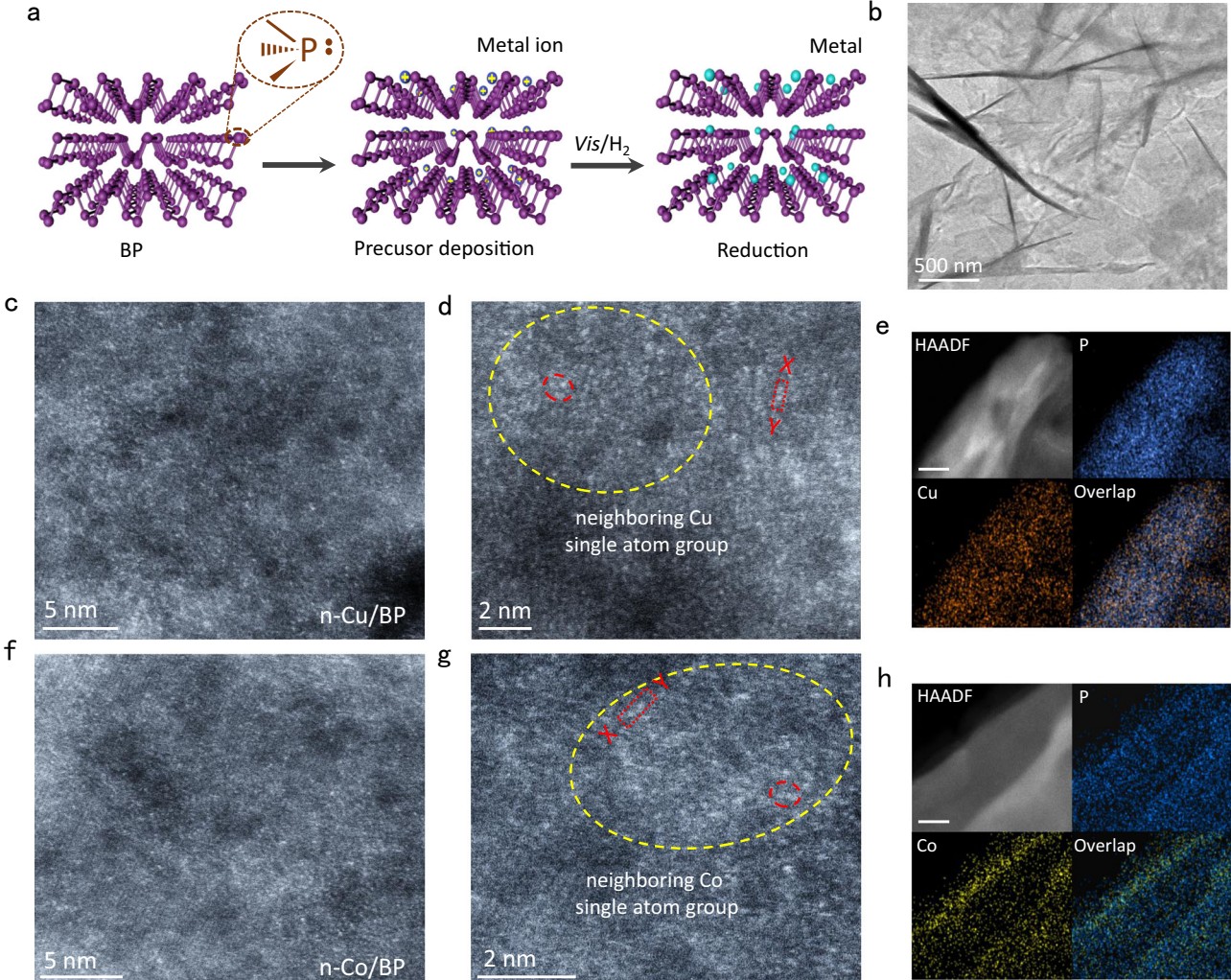

**Fig. 1 | Synthesis and characterization of the atomically dispersed n-Cu/BP and n-Co/BP catalysts. a** Schematic illustration of regulating the reactivity of BP when the lone-pair electrons of surface P atom coordinates with metal ions or single atoms (the atomic structure of BP was drawn by Cinema 4D software). **b** Representative TEM image of n-Cu/BP. **c, d** HAADF-STEM image of n-Cu/BP. **e** STEM-EDS elemental mapping of n-Cu/BP nanosheets. Scale bar, 20 nm. **f, g** HAADF-STEM image of n-Co/BP. **h** STEM-EDS elemental mapping of n-Co/BP nanosheets. Scale bar, 20 nm.

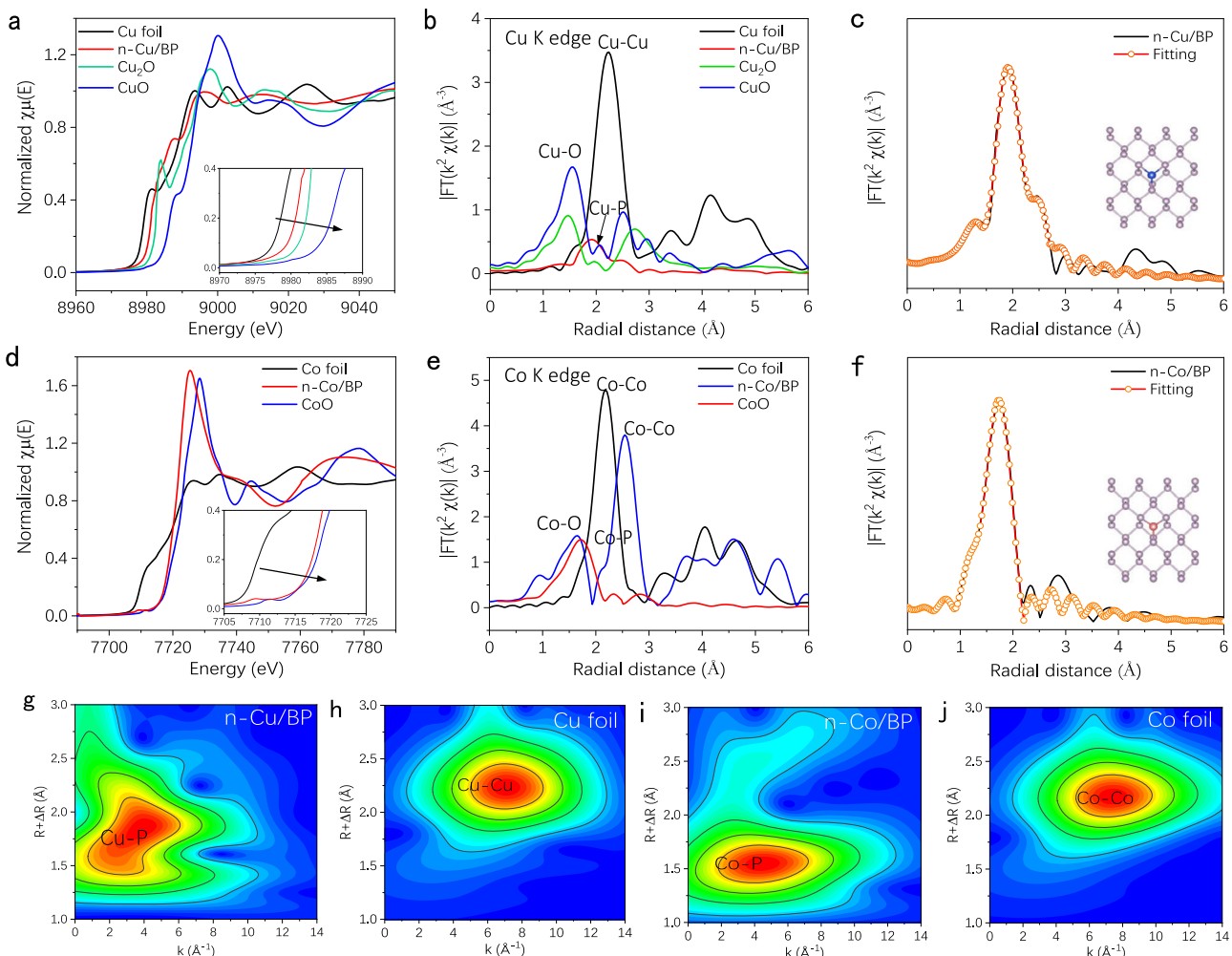

**Fig. 2 | Chemical state and coordination information for copper and cobalt SACs. a** The normalized Cu K-edge XANES and **b** FT-EXAFS spectra of n-Cu/BP, Cu foil, Cu₂O and CuO. **c** FT-EXAFS fitting curves of Cu K-edge for n-Cu/BP (inset: structural model of Cu-P₃). **d** The normalized Co K-edge XANES and **e** FT-EXAFS spectra of n-Co/BP, Co foil and CoO. **f** FT-EXAFS fitting curves of Co K-edge for n-Co/BP (inset: structural model of CoP₃). **g, h** WT contour plots of Cu K-edge at R space for n-Cu/BP and Cu foil. **i, j** Co K-edge at R space for n-Co/BP and Co foil.

in a neutral environment (Supplementary Fig. 4). Because of the strong ability of metal ions to accept electrons, they can interact with the partial electron of P atoms by electrostatic interaction[45,46]. Cu(Ac)₂·H₂O and Co(Ac)₂·4H₂O (Ac, acetate) were introduced into an NMP dispersion of BP to allow the adsorption of Cu and Co species, respectively. Inductively coupled plasma optical emission spectrometry (ICP-OES, Supplementary Figs. 5, 6) conforms to the high adsorption capacity of the metal ions on the BP surface, up to 30 wt%. The mixture was then photoinduced reduction assisted by hydrogen. After 3 h of visible light irradiation, the M/BP (Cu/BP and Co/BP) catalysts were collected and washed thoroughly with ethanol and water.

The X-ray diffraction (XRD) patterns of Cu/BP and Co/BP show no signals associated with crystalline Cu or Co species (Supplementary Fig. 7), revealing their high dispersion. M/BP nanosheets were observed along [200] and [002] crystallographic directions by the atomic-resolution scanning transmission electron microscopy (STEM) (Supplementary Fig. 8). Some dark atoms in the lattice can be distinguished as single metal atoms on the BP surface. High-angle annular dark-field scanning transmission electron microscopy (HAADF–STEM) images were allowed direct observation of a high density of isolated atoms uniformly dispersed on BP nanosheets (Fig. 1c, f). Numerous dispersed metal atoms constitute groups of neighboring atoms, forming the dense single-atom group structures (defined as n-M/BP, Fig. 1d, g and Supplementary Fig. 9). The line profiles for the HAADF

images along X–Y elucidate that the Cu/Co atoms are separated by at least 0.29 nm (Supplementary Fig. 10). Energy-dispersive x-ray spectroscopy (EDS) analysis in a STEM reveals that metal atoms are evenly dispersed in n-Cu/BP and n-Co/BP, respectively (Fig. 1e, h), unlike in supported Cu/Co nanoparticles prepared by a conventional thermal reduction method (Supplementary Fig. 11). The Cu and Co loadings in n-Cu/BP and n-Co/BP measured by ICP-OES are 11.3 wt% and 5.2 wt%, respectively. This metal loading is higher than that of the most of reported SACs (Supplementary Table 1). We further investigated the HAADF–STEM images of Cu/BP with different Cu mass loading in Supplementary Fig. 12 and Supplementary Table 2. Cu atoms in Cu₁.₅₂/BP are sparsely distributed as isolated atoms. As Cu loading increases to 3.93 wt%, a small fraction of Cu atoms form groups of neighboring atoms, while most remain as isolated atoms. At 15.8 wt%, most Cu atoms gradually aggregate to form nanoclusters, which is in visible contrast to the neighboring Cu single atom group.

The synchrotron radiation-based X-ray absorption fine structure spectroscopy (XAFS) results reveal the atomic structure and coordination state of the single-atom Cu and Co species in n-Cu/BP and n-Co/BP. The Cu K edge X-ray absorption near-edge structure (XANES) spectrum of n-Cu/BP compared with standard Cu foil, Cu₂O and CuO are shown in Fig. 2a. The absorption edge of n-Cu/BP is located between the Cu foil and Cu₂O, suggesting that the Cu atoms possess a slightly positive valence state loaded between 0 and +1, presumably

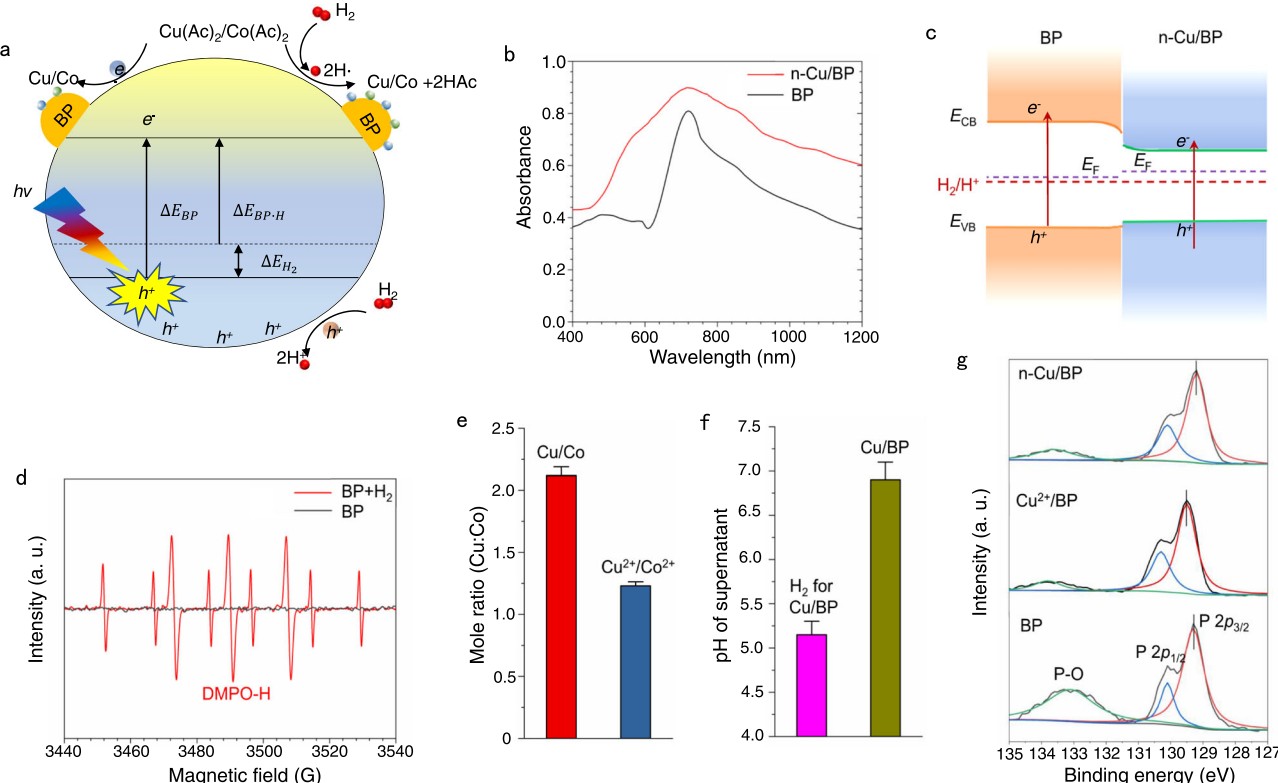

**Fig. 3 | Fundamental understanding of the synthesis process of electrocatalyst.**
**a** Schematic diagram of photochemical induced M/BP synthesis. **b** UV/Vis DRS spectra. **c** Energy diagram of BP and n-Cu/BP, respectively ($E_F$ respects Fermi level). **d** ESR spectra of the solutions obtained after 150 s of visible-light-induced reaction for $H_2$ decomposition on BP. **e** Molar ratios of Cu to Co in M/BP (Cu/BP and Co/BP) and $M^{2+}$/BP ($Cu^{2+}$/BP and $Co^{2+}$/BP), and box plots indicate median. **f** Change in pH after $H_2$ was introduced into the NMP dispersion of $Cu(Ac)_2$ and BP, and box plots indicate median. **g** P 2*p* XPS spectra of n-Cu/BP, $Cu^{2+}$/BP, and BP.

attributed to the fact that Cu and P have a similar electronegativity. In contrast, XANES of Co K edge in n-Co/BP reveals a high average valence close to +2 in Fig. 2d. Such a different oxidation state between Cu and Co is ascribed to higher reducibility of Co atoms. The coordination environment of Cu was confirmed by the extended XAFS (EXAFS, Fig. 2b). The Fourier-transformed (FT) $k^2$-weighted EXAFS of Cu in n-Cu/BP shows a main peak at around 1.92 Å, corresponding to the first coordination shell Cu−P coordination. There is a slight peak at around 2.44 Å belonging to the Cu-Cu bond, indicating that the dense neighboring Cu atoms exist weak interaction. By contrast, n-Co/BP shows a dominant peak around 1.71 Å, different from that of the Co−O bond (1.60 Å) in CoO and Co−Co bond (2.60 Å) in Co foils (Fig. 2e), which can be attributed to the existence of Co−P bonds. With the further increase of Cu atom loading, $Cu_{15.8}$/BP displays an additional minor peak of Cu−Cu scattering at 2.4 Å (Supplementary Fig. 13). By contrast, no obvious peaks at 2.4 Å for $Cu_{11.3}$/BP and $Cu_{3.93}$/BP evidence the atomical dispersion of Cu atoms, in accordance with the HAADF-STEM observations.

EXAFS fitting of n-Cu/BP and n-Co/BP were performed at the Cu K-edge and Co K-edge to extract the structural parameters (Supplementary Table 3). The EXAFS fitting curve of Cu is given in Fig. 2c. The Cu atom is connected by three P atoms at the first coordination shell (namely $CuP_3$ coordination), with a bond length of 2.34 Å. The fitting EXAFS spectrum in Fig. 2f shows the $CoP_3$ coordination in n-Co/BP with a Co-P bond length of 2.16 Å. The fitting results indicate that all atomic Cu and Co sites are three-coordinated by phosphorus species and the atomic structure models are illustrated in Fig. 2c, f (inset), respectively. The wavelet transform (WT) for the $k^2$-weighted EXAFS signals of n-M/BP and corresponding contrast materials were carried on the complex wavelet developed by Morlet (Fig. 2g–j and Supplementary Fig. 14). The WT contour plots of Cu and Co in n-Cu/BP and n-Co/BP show one

intensity maximum at 3.9 and 4.2 Å$^{-1}$, respectively, ascribed to Cu-P and Co-P bonds by comparison with those of contrast materials. Moreover, the XANES spectra of Cu in $Cu^{2+}$/BP and Co in $Co^{2+}$/BP were further studied (Supplementary Figs. 15 and 16). Both valence states of copper and cobalt are close to +2. The main peak of Cu FT-EXAFS at around 1.41 Å is ascribed to Cu-O bonds, and a weak peak at 1.93 Å belongs to the Cu-P bond, which indicates adsorption of $Cu(AC)_2$ species bound on BP surface by the weak interaction. There are no Cu−Cu bonds in $Cu^{2+}$/BP, implying that $Cu(AC)_2$ species are evenly distributed. In EXAFS spectra of Co for $Co^{2+}$/BP, unlike that of Co/BP, a peak in the region 2−3 Å from the Co−Co contribution, which may be attributed to the easier aggregation of adsorbed $Co(AC)_2$ during the removal of organic residues on the surface of $Co^{2+}$/BP by calcination.

**Fundamental understanding of the synthesis process of M/BP**
To better understand why the M/BP (Cu/BP as the main research) catalysts possessed such a stable and high loading, we further explained the synthesis process, as seen in the schematic diagram of photochemical induced process (Fig. 3a). BP has the adjustable direct-band-gap properties, enabling it to work as an efficient photocatalyst with broadband solar absorption[47]. Under visible light irradiation, the charges are generated to drive the in-situ reduction of metal ions adsorbed on the BP surface. We synthesized Cu/BP nanosheets under vis irradiation, as seen in the illustration in Supplementary Fig. 17. However, a low and sparse loading of Cu single atoms on BP support was shown by the corresponding HAADF−STEM images and STEM-EDS elemental mapping (Supplementary Fig. 18). This may be attributed to the high oxidation potential of organic anions as electron donors, which is difficult to be directly or indirectly oxidized by photo-generated holes, resulting in the rapid recombination of photo-generated carriers, thereby affecting the photochemical reduction

process. We thus introduced hydrogen into the catalytic reaction system as a hole-trapping agent to inhibit the recombination of carriers. The ICP-OES results show a significantly increased single-atom loading with the addition of $H_2$ (Supplementary Table 1 and 4), which implies that injecting $H_2$ could effectively accelerate the kinetic process of metal ion reduction. To rule out the possible reduction effect of hydrogen in the reaction liquid, we further put the mixture in the dark and pumped $H_2$ for 3 h continuously (Supplementary Fig. 19). There were no Cu and Co species detected in the substrate by ICP-OES, confirming that Cu and Co ions could not be reduced into single atoms or nanoparticles in liquid phase at room temperature.

Appropriately aligned band structures are critical to achieve photoreduction of metal ions. The light-harvesting capability of the BP and Cu/BP was investigated by UV/Vis diffuse reflectance spectra (DRS). Cu/BP display a wider absorption from the Vis to infrared (IR) region (Fig. 3b). The band gap values ($E_g$) of BP and Cu/BP were estimated to be about 1.17 and 1.06 eV, respectively (Supplementary Fig. 20). The ultraviolet photoelectron spectra (UPS) are shown in Supplementary Fig. 21. The maximum valence band values ($E_{VB}$) of BP and Cu/BP could be calculated to be 0.72 and 0.7 eV (vs. normal hydrogen electrode (NHE)), respectively. The corresponding energy level diagrams of BP and Cu/BP are shown in Fig. 3c and Supplementary Fig. 22. It can be seen that the reduction potential of Cu/Co metal ions is higher than the conduction band edge of BP, so the photocatalytic reduction process can proceed in thermodynamics. The reduced band-gap in Cu/BP is conducive to the transition of electrons. At the same time, the load of metal atoms can capture photogenerated electrons, thus inhibiting the recombination of charges and carriers, and further improving the photoreduction rate[48–50].

To further verify the role of hydrogen in the synthesis of catalyst, we used 5,5-dimethyl-1-pyrroline-N-oxide (DMPO) as the radical trapping reagent to detect the existence of hydrogen radicals (H·). As $H_2$ was injected into the reaction system, nine peaks of electron spin resonance (ESR) were observed when the product from the reaction was added to DMPO (Fig. 3d)[51,52]. The signal could be assigned to a spin adduct of DMPO-H, which implies that $H_2$ dissociates into H· on BP surface under visible light irradiation and then involves in the redox process. In contrast, there is no obvious signal peaks in the BP without $H_2$. With the assistance of $H_2$, the loading of Cu/Co atoms is significantly increased. But their atomic ratios are different before and after the reaction. The atomic ratio of Cu to Co increases from 1.3:1 to 2.2:1 (Fig. 3e), mainly due to the higher reduction potential of Cu than Co. When $Cu(Ac)_2$ and $Co(Ac)_2$ were both used as reactants for photoinduced reduction reaction, the bi-atomic CuCo/BP catalyst was synthesized with a close Cu/Co atomic ratio of 2.4:1 (Supplementary Fig. 23 and Supplementary Table 5). HAADF-STEM and STEM-EDS images in Supplementary Fig. 24 reveal the presence of Cu and Co atoms with a dense and random distribution on or near the surface of BP. The pH values of mixtures with and without $H_2$ treatment were further evaluated by detecting supernatants after photoreduction in Fig. 3f. A decrease of pH from 6.7 to 5.3 after $H_2$ treatment is attributed to the formation of $H^+$. Moreover, the lower pH also inhibits the adsorption of $Cu^{2+}/Co^{2+}$ on BP surface according to zeta-potential of BP in NMP, which to some extent avoids the single-atomic agglomeration caused by excessive reduction of metal ions.

X-ray photoelectron spectroscopy (XPS) was further applied to gain insight into the valence states of M/BP. Compared with BP nanosheets in Fig. 3g, the P $2p_{3/2}$ peak of Cu/BP shows a small shift to low energy, indicating a slight electron-rich state of P atom. In $Cu^{2+}/BP$, the slight positive shift of P $2p_{3/2}$ peak is attributed to the electrostatic adsorption of metal ions. Moreover, the intensity of P–O bonds in Cu/BP and $Cu^{2+}/BP$ is lower than that of in BP, confirming its low absorbability to oxygen, thus enhancing the structural stability of BP[45,46]. In addition, the Cu $2p_{3/2}$ and Co $2p_{3/2}$ XPS peaks in M/BP shift by −0.8 and −0.4 eV compared with that of in $M^{2+}/BP$, respectively, which suggests

that Cu and Co ions are successfully reduced to low valence state of metal atoms (Supplementary Fig. 25).

## Electrocatalysis of alkaline HER

We compared the electrocatalytic HER activities of n-Cu/BP catalysts by linear sweep voltammetry (LSV) scans, using a three-electrode configuration with simultaneous iR correction. For comparison, n-Co/BP, CuCo/BP (Cu and Co loadings are 8.8 and 3.7 wt%), commercial Pt/C and BP nanosheets were also investigated under the same conditions. In Fig. 4a, single-atom n-Cu/BP shows a low overpotential of 41 mV at 10 mA cm$^{-2}$, which is close to that of Pt (39 mV) and 100 mV lower than that of n-Co/BP. At a higher potential range, the electrocatalytic activity of n-Cu/BP even surpasses Pt/C, demonstrating that Cu single atoms stabilized on BP nanosheets can significantly improve the electrocatalytic performance. The corresponding Tafel slopes are further analyzed as shown in Fig. 4b. The n-Cu/BP sample exhibits a Tafel slope of 53.4 mV dec$^{-1}$, similar to commercial Pt/C (58.8 mV dec$^{-1}$) and much lower than that of n-Co/BP (131.6 mV dec$^{-1}$), indicating faster the Volmer–Heyrovsky kinetics. The n-Cu/BP exhibits superior HER activity than bi-atomic CuCo/BP, which could explain the critical role of adjacent Cu atoms for enhancing reactivity. Moreover, the exchange current density ($j_0$) determined by extrapolating the Tafel plot of n-Cu/BP is estimated at 1.91 mA cm$^{-2}$ (Supplementary Fig. 26), revealing the excellent inherent electrocatalytic activity than those of other catalysts. We further assess the HER activity of $Cu^{2+}/BP$ and Cu NPs/BP in Supplementary Fig. 27. The metal ions in oxidation state or nanoparticles on BP present a distinctly sluggish electrocatalytic HER, suggesting that the high HER performances of n-Cu/BP are contributed from atomically dispersed Cu sites with Cu-$P_3$ coordination. Therefore, n-Cu/BP exhibits desirable HER performance compared with recently reported non-noble metal-based catalysts (Supplementary Table 6).

The turnover frequency (TOF) (Fig. 4c) and mass activities (Fig. 4d) are calculated to investigate the intrinsic hydrogen-evolving activity of BP-supported neighboring Cu catalytic sites on the basis of ICP−OES results. The TOF value of n-Cu/BP at an overpotential of 150 mV was calculated to be 0.53 $H_2$ s$^{-1}$, which was three times higher than that of bi-atomic CuCo/BP (0.17 $H_2$ s$^{-1}$) and four times that of n-Co/BP (0.12 $H_2$ s$^{-1}$). A detailed comparison of TOF values (Supplementary Table 7) shows the high activity of our n-Cu/BP. n-Co/BP exhibits a specific current of 0.82 A mg$^{-1}$ at 100 mV. When increasing the overpotential to 300 mV, the specific current values increase to 5.49 A mg$^{-1}$, which is more than four times the values of CuCo/BP or n-Co/BP. These results indicate that neighboring single-atom Cu sites possess better catalytic kinetics than neighboring Co or CuCo sites on BP surface.

Figure 4e and Supplementary Fig. 28 show the kinetic activities of Cu/BP and Co/BP with different metal loading, respectively. The overpotential dramatically decreases at higher Cu or Co loading, while the lower activity of Cu$_{15.8}$/BP compared with Cu$_{11.3}$/BP is mainly attributed to the formation of nanoclusters by single atom aggregation. A positive correlation of composition between neighboring Cu or Co sites suggests that the activity of n-Cu or n-Co is much higher than that of isolated single atoms. In addition, we kept the feeding amount of Co at 0.05 mmol and changed the Cu amount from 0 to 0.1 mmol in the synthesis of bi-atomic CuCo/BP (Supplementary Table 8). As seen in Fig. 4f and Supplementary Fig. 29, we fixed the feeding amount of Cu to 0.05 mmol. The overpotential increases along with the increase of Co ratio and reaches the highest overpotential of 101 mV for Cu$_{0.05}$Co$_{0.1}$/BP. Moreover, the overpotential at 10 mA cm$^{-2}$ decreases from 141 mV (Cu$_0$Co$_{0.05}$/BP) to 113 mV (Cu$_{0.025}$Co$_{0.05}$/BP). With a further increase of Cu in the system, the overpotential reaches only 73 mV (Cu$_{0.05}$Co$_{0.05}$/BP). When further increasing the Cu amount to 0.1 mmol for Cu$_{0.1}$Co$_{0.05}$/BP, the overpotential then rises slightly again to 76 mV. This activity trend shows that properly increasing the Cu single atom loading can significantly improve the reactivity.

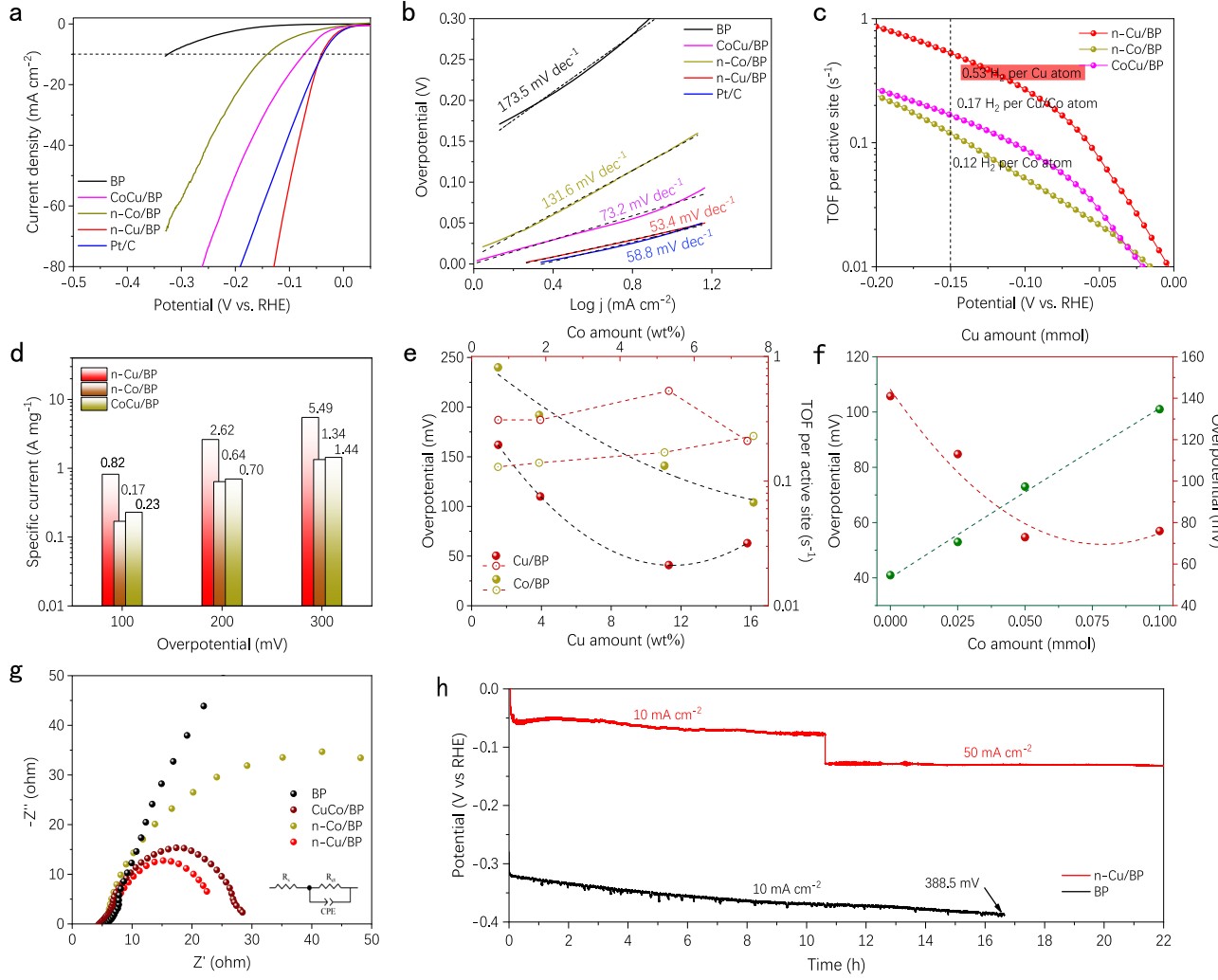

**Fig. 4 | HER performance of n-Cu/BP catalysts and control samples. a** LSV curves and **b** Tafel plots of the catalysts in 1 M KOH with a scan rate of 5 mVs⁻¹. **c** TOFs of Cu, Co, and CuCo atoms on BP at different overpotentials. **d** Mass activities of the catalysts on the basis of Cu, Co and CuCo atoms. **e** Overpotentials of Cu/BP and Co/BP with different Cu and Co loading, respectively, and the corresponding TOFs activity trend. **f** Overpotentials of CuCo/BP with different CuCo ratios (fixed the feeding amount of Cu or Co at 0.05 mmol and change in Co or Cu amount), and the corresponding activity trend. **g** Nyquist plots of experimental data for BP, CuCo/BP, n-Co/BP, and n-Cu/BP. **h** Stability of n-Cu/BP and BP catalysts.

The electrochemical impedance spectroscopy (EIS) based on an equivalent circuit model was carried out to investigate the charge-transport properties in Fig. 4g. n-Cu/BP delivers significantly lower charge transfer resistance ($R_{ct} = 16.7\ \Omega$) than that of CuCo/BP (22.3 Ω) and n-Co/BP (78.4 Ω), suggesting the more favorable absorption of the hydrogen intermediates and a fast Faradaic reaction process. The electrochemical active surface area (ECSA) of each as-prepared catalysts was estimated by determining the double-layer capacitance ($C_{dl}$) (Supplementary Fig. 30). The n-Cu/BP is found to possess a larger $C_{dl}$ (49.0 mF cm⁻²) than CuCo/BP (37.9 mF cm⁻²) and n-Co/BP (22.3 mF cm⁻²), indicating more accessible active sites from the neighboring Cu atoms in the BP basal planes. The ECSA-normalized current density of n-Cu/BP still performs the highest intrinsic activity in Supplementary Fig. 31. This result indicates that the HER intrinsic activity depends crucially on the synergistic effect of neighboring bi-Cu metal sites.

The catalytic stability is essential to evaluate the device application potential of HER catalysts. The stability of n-Cu/BP was estimated by chronoamperometric test under a constant current of 10 mA cm⁻² and 50 mA cm⁻² in Fig. 4h, showing a stable overpotential for 22 h. In contrast, When the electrode works at 10 mA cm⁻² (current provided from BP was subtracted) for only 16.6 hours, a rapid increase in

overpotential of 74 mV was observed, which was mainly ascribed to the easily oxidized surface of BP by oxygen in water when no metal atoms was loaded in BP[46,53,54]. Moreover, after 2500 cyclic-voltammetry (CV) cycles, the LSV curves and corresponding mass activities of n-Cu/BP at different overpotential were presented in Supplementary Figs. 32 and 33. n-Cu/BP still retains 90% of its original electrocatalytic activity, indicating its considerable electrochemical stability. The HAADF-STEM images and element mapping of n-Cu/BP after long-term electrocatalysis are shown in Supplementary Figs. 34 and 35. The dense single-atom group is still presented without any aggregations for n-Cu/BP catalyst. After long-time operation, the XANES and FT-EXAFS spectra of n-Cu/BP shows that the single-atom Cu sites remain atomic dispersion without aggregation (Supplementary Fig. 36). The Raman shifts of n-Cu/BP and BP nanosheets after 2500 CV cycles are collected in Supplementary Fig. 37. n-Cu/BP still maintains these three typical peaks corresponding to Raman spectra of BP. The pure BP is transformed into red phosphorus or phosphorus oxide ($PO_x$) after long-time electrocatalysis[47,55]. As evidenced by XPS characterizations in Supplementary Figs. 38 and 39, only a little change in the composition occurred for the P−O and Co−O bonds in n-Cu/BP after long-term CV cycles, which is ascribed to the existence of slight oxidation or OH⁻ absorption. The faradaic efficiency (FE) for HER was determined by

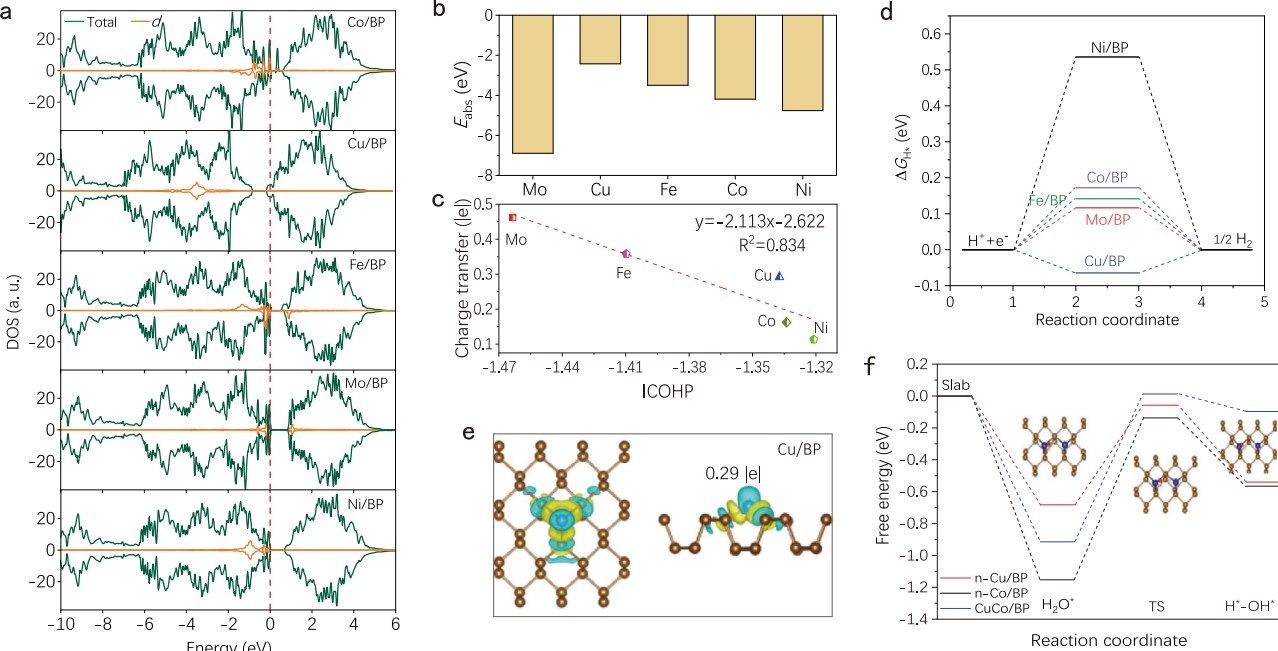

**Fig. 5 | Theoretical calculations of n-Cu/BP on HER. a** Total and partial DOS of Cu, Co, Fe, Ni, and Mo single atoms on BP support. **b** The adsorption energy values of the common transition metal single-atom models. **c** The theoretically calculated charge transfer values for metal single atoms on BP as a function of the ICOHP values. **d** Calculated free energy diagram of the HER with M-P₃ as an active center. **e** The charge density difference diagram of Cu/BP and corresponding charge transfer from the Cu atom to the BP layer. Cyan and yellow contours represent electron depletion and accumulation. The isosurface value is 0.002 e/Å³. **f** Free energy of water dissociation diagram for n-Cu/BP, n-Co/BP and CuCo/BP.

comparing the amount of measured gas with theoretically calculated gas. The accordant values suggest that the FE is close to 100% (Supplementary Fig. 40).

## Theoretical study of n-Cu/BP on HER

We introduced transition metal atoms to decorate BP nanosheets to screen only the SACs with well-balanced empty/occupied d orbitals, activating the $H_2$ molecules. Figure 5a displays the transition metals (Mo, Cu, Fe, Co, and Ni) on BP support of density of state (DOS). The Fermi level crosses the conduction band of Cu/BP, exhibiting metallic properties. This phenomenon indicates higher conductivity in Cu/BP structure, affecting favorably the electrocatalytic HER[56,57]. The Cu 3d state is located in a more negative region with the most negative d-band center (−3.51 eV) than those in other transition metals on BP, suggesting that the Cu sites show the more enriched electron state on Cu/BP[42,58]. The single-atom absorption energies (ΔE) were calculated subsequently (Fig. 5b). Although the Cu atom absorbed on BP surface possesses strong bonding with a high ΔE value (−2.43 eV), it is a relatively weak interaction compared with other candidates. The charge density analysis in Fig. 5c, e shows that the incorporation of heteroatoms has an appreciable influence on electron distribution. After coupling single atoms with BP, the charge density in the hybrid's interlayer is redistributed in the form of an apparent electron transfer from those metal atoms to BP. Moreover, we extracted the bonding states of the H atom adsorbed on M-P₃ structures by crystal orbital Hamilton population (COHP) analysis in Supplementary Fig. 41. The negative COHP represents the bonding contribution and the positive COHP stands for the antibonding contribution. A more negative ICOHP value is responsible for a stronger interaction[59]. Notably, as the positive charge of these transition metal atoms increases in turn, their bonding energy with H* becomes stronger. The final charge transfer value as a function of the ICOHP for these metal SACs presents a well linear relation. Single-atom Cu site has a moderate M-H interaction and shows the positive charge of 0.29. The free energies of hydrogen adsorption (ΔG_{H*}) on the surface of BP nanosheet-supported single-

atom catalysts (Cu, Co, Fe, Ni, and Mo) are calculated in Fig. 5d. Single-atom Cu/BP catalyst shows a most optimal ΔG_{H*} value of −0.06 eV, which is closer to the ideal value (i.e., 0 eV). By compared from recently published studies on electron-defect-/-rich single-atom Cu catalysts with different coordination structures in Supplementary Table 9, the largest bond length of Cu−P and lowest charge transfer value for n-Cu/BP catalyst demonstrates its weak non-strong interaction between Cu and BP as well as electron-rich properties of Cu sites.

The thermal stability of single-atom Cu/BP catalyst was surveyed by ab initio molecular dynamics (AIMD) simulations at 500 K for 10 ps with a time step of 2 fs in Supplementary Fig. 42. The total energy oscillates near the initial condition, and although the structure of Cu/BP is wrinkled, the single-atom Cu is still embedded in the BP monolayer, which illustrates the thermodynamic stability of Cu/BP.

The key reaction steps for destabilizing water in alkaline HER on single atom Cu sites were investigated. The atomic model of isolated Cu single atoms on BP surface was simulated by the climbing-image nudged-elastic-band method (CI-NEB) in Supplementary Fig. 43. $H_2O$ molecule adsorbs onto the Cu sites, then dissociates into adsorbed H⁻ and OH⁻ with the help of free electrons on the Cu and nearby P atoms of BP, respectively. However, the corresponding CI-NEB result shows a high barrier from initial state (IS) configuration to the final state (FS) configuration. The unstable configuration of final state is mainly due to the weak H*-adsorption energy for P atoms, which makes it difficult for H* to transfer to nearby P atom. Therefore, the isolated Cu single-atom sites on BP surface possess low HER activity, which is consistent with experimental results.

The reaction path of water dissociation for two neighboring Cu atoms was further investigated based on the NEB method. As illustrated in Fig. 5f, when two neighboring Cu atoms act as adsorption sites of H⁻ and OH⁻, respectively, the activation energy of n-Cu/BP for the dissociation of the water molecule is 0.63 eV, lower than that of the isolated Cu/BP (1.82 eV). This indicates that O−H bond is prone to fracture, leading to a faster H⁻ supply, which accelerates the slow Volmer step. Likewise, the atomic models of n-Co/BP and bi-atomic

CuCo/BP were further investigated by CI-NEB. As Cu atom are replaced by Co atom, the corresponding energy barriers of water dissociation is gradually increased. Our DFT calculations confirm that the high activity of n-Cu/BP for HER arises from two neighboring Cu atoms because of the nearly zero Gibbs free energy and preferable water dissociation activity.

## Discussion

In this work, we report a room-temperature photochemical strategy with hydrogen auxiliary to produce the stable, high-loading SACs (Cu, Co) on BP nanosheets. Visible light-induced formation of hydrogen radicals (H) on BP nanosheets is shown to be critical for preparing high-loading n-Cu/BP with metal-atomic loading up to 11.3% (the loading of Co is 5.2% for n-Co/BP). The obtained n-M/BP catalysts show an exceptional electron-rich feature, which increases their $H_2$-production activity in alkaline HER. The results indicate that the active sites of single-atom Cu and Co both show the $\triangle G_{H^*}$ close to zero for hydrogen evolution. A neighboring Cu SAC exhibits the highest water dissociation activity, significantly surpassing both isolated Cu SAC and Cu nanoclusters. In addition, n-Cu/BP still shows a greater turnover frequency (TOF) than that of surface neighboring Co SAC (n-Cu/BP) and bi-atomic CuCo/BP, when Co atom is introduced. This work reveals that there is still plenty of room for catalyst design in isolated single atoms to obtain higher activity.

## Methods

### Synthesis of 11.3 wt% n-Cu/BP

Black phosphorus (0.1 mmol) was added into NMP solution (30 mL) with ultrasonic exfoliation for 30 min at Ar atmosphere. Then Cu(Ac)$_2$·H$_2$O (0.05 mmol) was dissolved in the mixture solution with stirring for 2 h. The mixed metal ion solution was treated by photoinduction method under flowing $H_2$ (flow rate: 40 mL min$^{-1}$). A 300 W Xe arc lam was used as a visible light source (a cut-off filter, $\lambda > 400$ nm). After 3 h irradiation, the suspension was collected and washed by ethanol and cyclohexane (4:1), and then dried at room temperature for 12 h in a vacuum.

### Synthesis of 5.2 wt% n-Co/BP

Co(Ac)$_2$·4H$_2$O (0.05 mmol) was dissolved in the mixture solution with stirring for 2 h. The mixed metal ion solution was also treated by photoinduction method under flowing $H_2$ (flow rate: 40 mL min$^{-1}$) for 3 h irradiation. Then the suspension was filtered, washed, and collected.

### Synthesis of bi-atomic CuCo/BP (Cu 8.8 wt% and Co 3.7 wt%)

Cu(Ac)$_2$·H$_2$O (0.05 mmol) and Co(Ac)$_2$·4H$_2$O (0.05 mmol) were both dissolved in the mixture solution with stirring for 2 h. The mixed solution of Cu and Co ions was treated by photoinduction method under flowing $H_2$ (flow rate: 40 mL min$^{-1}$) for 3 h irradiation. Then the suspension was filtered, washed, and collected.

### Synthesis of Cu NPs/BP and Co NPs/BP

Cu(Ac)$_2$·H$_2$O (0.05 mmol) and Co(Ac)$_2$·4H$_2$O (0.05 mmol) were dissolved in the mixture solution with stirring for 2 h, respectively. Then the suspension including Cu or Co metal ion was filtered, washed, and collected. The collected samples were heated at 500 °C at Ar atmosphere for 2 h, respectively.

### XAFS experiment

The XAFS analysis of Cu K-edge and Co K-edge were obtained at the 1W1B station of Shanghai Synchrotron Radiation Facility (SSRF), which was operated at 2.5 GeV with a maximum current of 250 mA. The obtained EXAFS results were processed on the basis of the standard procedures using the ATHENA module implemented in the IFEFFIT software packages.

## Electrochemical measurements

All electrochemical measurements were performed with a three-electrode system using CHI 760E electrochemical work station under environmental chamber (CH Istruments, Inc., Shanghai). The glassy carbon electrode (GCE, diameter = 3 mm at a catalyst loading of 0.57 mg cm$^{-2}$), Hg/HgO electrode, and carbon rod were worked as the working electrode, reference electrode, and counter electrode, respectively. And the electrolyte was saturated with hydrogen prior to the electrochemical test. The electrocatalytic activity of all the samples towards HER was tested by obtaining polarization curves using linear sweep voltammetry (LSV) at the voltage range from +0.15 to −0.44 V (vs. RHE) with a scan rate of 5 mV s$^{-1}$ in 1 M KOH solution, respectively. The polarization curves were recorded by 95% IR compensation. The double-layer capacitances ($C_{dl}$) were estimated by CV at various scan rates (20–120 mV s$^{-1}$) to evaluate the effective surface area of various catalysts. By plotting the $\Delta J = (J_a - J_c)$ at 0.14 V versus RHE against the scan rate, the linear slope that is twice of the double layer capacitance ($C_{dl}$) is used to represent ECSA.

## Computational methods

In this simulation, all DFT calculations were carried out with the Vienna ab initio Simulation Package (VASP)[60]. The electro-ion interactions was described by the projector-augmented wave (PAW) pseudopotentials[61]. The generalized gradient approximation (GGA) was used with the Perdew-Burke-Ernzerhof (PBE) exchange-correlation functional[62]. The DFT-D3 method was adopted to treat van der Waals (vdW) interactions in the systems[63]. The kinetic energy cut-off was set 450 eV in all computations to describe all atoms' valence electrons. In all calculations, the convergence criterion for minimum energy and minimum force during geometrical optimization was used as $10^{-5}$ eV and 0.03 eV/Å, respectively. The theoretically optimized lattice constant of BP ($a = 3.31$ Å, $b = 4.38$ Å, $c = 10.50$ Å) agree well with the experimental value ($a = 3.31$ Å, $b = 4.38$ Å, $c = 10.48$ Å), indicating the feasibility of our calculation approaches for modeling this system. A $3 \times 3 \times 1$ monolayer of was built with a sufficient vacuum gap of 15 Å to prevent the interaction between the periodic images. The Brillouin zone was sampled using a $3 \times 3 \times 1$ grid centered at the gamma ($\Gamma$) point for fully relaxed geometry optimization, while a $6 \times 6 \times 1$ k-points grid was employed for electronic property computations. Ab initio molecular dynamics (AIMD)[64] simulation was employed to evaluate the stability of Cu/BP under the NVT ensemble ($T = 300$) and all simulation times were 10 ps with a time step of 2 fs. The Bader charge analysis was carried out to obtain the amount of charge transfer[65]. The integrated-crystal orbital Hamilton population (ICOHP) was obtained by calculating the band states integral up to the highest occupied energy level[66]. The spin-polarized computation was performed. The climbing image nudged elastic band (CI-NEB) method[67] was applied to compute the decomposition barriers of $H_2O$ molecular to obtain the minimum energy path between the given initial and final positions. The Gibbs free energy $\triangle G_{H^*}$ is defined as follows:

$$\triangle G_{H^*} = \triangle E_{H^*} + \triangle ZPE - T\Delta S \qquad (1)$$

Where $\triangle E_{H^*}$ is the hydrogen chemisorption energy, $\Delta ZPE$ is the zero-point energy difference between absorbed and the gas phase, and $T\Delta S$ is the entropy change of $H^*$.

The adsorption energy ($E_{ads}$) of metal single atoms was calculated as follows:

$$E_{abs} = E_{total}(M + substrate) - E_{total}(M) - E_{total}(substrate) \qquad (2)$$

in which $E_{total}(M + substrate)$, $E_{total}(M)$, and $E_{total}(substrate)$ are the total energy of BP with adsorbed metal single atom, metal single atom, and BP substrate, respectively.

## Data availability

The data that support the findings of this study are available from the corresponding authors upon reasonable request. Source data are provided as a Source data file. Source data are provided with this paper.

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

## Acknowledgements

This work was financially supported by the Fundamental Research Funds for the Central Universities (0301005202017, 2018CDQYFXCS0017, 106112017CDJXSYY0001), Thousand Young Talents Program of the Chinese Central Government (Grant No. 0220002102003), National Natural Science Foundation of China NSFC, Grant No. U19A20100, 21971027, 21373280, 21403019), Beijing National Laboratory for Molecular Sciences (BNLMS) and Hundred Talents Program at Chongqing University (Grant No. 0903005203205), The Skate Key Laboratory of Mechanical Transmission Project (SKLMT-ZZKT–2017M11), Natural Science Foundation of Chongqing (Grant No. cstc2019jcyj-msxmX0426), Science and Technology Research Project of Education Agency in Chongqing (Grant No. KJZD-K201800102).

## Author contributions

Y.W. conceived and supervised the research; W.F. synthesized the catalysts and conducted performance test; J.W. and Y.L. performed DFT calculations. Y.W., H.Z. and W.F. analyzed the data and wrote the paper. J. L., W.C., and Z.G. discussed the results and commented on the manuscript.

## Competing interests

The authors declare no competing interests.
