## [Peer Review File · Nature Communications]

Photoinduced loading of electron-rich Cu single atoms by moderate coordination for hydrogen evolutionREVIEWER COMMENTS

Reviewer #1 (Remarks to the Author):

The manuscript focuses on single atom catalyst synthesized from photochemical methods on a black phosphorous support, which results into high catalyst loading. They used different characterization methods including imaging techniques, high energy x-ray methods and inductively coupled plasma spectrometry to confirm the presence of the single atoms. There are a few concerns that need to be addressed before this work can be accepted.

- 1) On their synthesis scheme (Figure 3a), it is claimed that H₂ plays a role as both a hole scavenger to inhibit quick charge recombination and also as an oxidant to the precursor. Is this process only selective to H₂? Did the authors try other hole scavenging chemicals and also rule out the formation of M-H structures given they used these catalyst for HER?
- 2) On Figure S11, the difference between single atom group and metal cluster should be correlated with EXAFS, where in the metal cluster case, the clear M-M bond is expected.
- 3) From their EXAFs analysis Co oxidation state is +2, does this imply their photochemical method does not reduce Co as compared to Cu?
- 4) What was the rationale chosen to select a cutoff energy of 350eV and the k-point densities for optimization and electronic properties? Were these values were properly converged, if so, to what tolerance

Reviewer #2 (Remarks to the Author):

In this work, authors have reported Cu P3 single-atom catalysts non-strongly coordinated on BP support at mild temperatures. This material was prepared at a high loading of more than 11%, of Cu loading. The prepared catalyst is well characterized by several techniques including TEM, STEM, HAADF-STEM, FT-EXAFS and XANEs, and other important characterization. Also, DFT studies have been explored in detail to prove the experimental investigations.

Some important comments:

1. The term "high electronegative ligand" is not clear at all. It is not clear, why the author calls it "high electronegative L" with respective what? What is the logic behind this design?
2. What about the reusability of these catalysts? any structural changes post reusability by XANES and HAADF-STEM to prove that the nature of SACs is stable?
3. Though, the author mentioned that high TOF, how it can be compared to previous procedures? A separate comparison table should be included in the SI. There are several easier procedures that are already reported with better results, as this preparation procedure is quite complex.
4. Some previous literature on Cu/ Co SACs and others should be included as respective section, e.g Chemical Reviews, 2021, 121,13620–13697; Small, 2021, 2006477; Coordination Chemistry Reviews Volume 418, 2020, 213376; <https://doi.org/10.1093/nsr/nwy077>; Small, 2021, 2006477; Advanced Materials Interfaces, 2021, 2001822; <https://doi.org/10.1021/acsnm.1c02743>.
5. what about the practicability of these catalysts?

Reviewer #3 (Remarks to the Author):

Authors of this work present a combined experimental and computational study on the preparation and characterization of single-atom catalysts on exfoliated black phosphorous flakes with ultrahigh atomic loading. Hydrogen was used as a hole-trapping agent for the preparation of SACs, which can improve the atomic loading and inhibit the aggregation of metal atoms. The authors show that the as-prepared SACs exhibit excellent HER activities. Here, I will mainly comment on the computational part of the manuscript, for which I have the following major concerns,

- 1) First, it is not clear why the authors choose and compare elements of Cu, Co, Fe, Ni, Mo, and Pt,

since there are so many metal atoms in the periodical table. It is not clear why the low binding energy of metal atom is related to the free energies of hydrogen adsorption. What is "obvious up-shifting Fermi level" (compared to what?) Meanwhile, it is not clear how this binding energy is defined. I would suggest "adsorption energy" from the calculated results, as all data are negative.

2) The authors claim "non-strong coordination", and there are quite large binding energies. Is this a contradictory? In literature, it was revealed that there are strong covalent bonding between Cu/Co and BP. Please refer to *Adv. Mater.* 2021, 33, 2008471. Both the metal atoms and BP flakes are stable after 2500 cycles of HER.

3) The authors should know the Fermi level is different for different systems. So have they considered the alignment of all energies to Vacuum level? If not, the Fermi level plots (Figs. 5a and 5b) will be meaningless.

4) The authors highlight that the metal atoms are electron rich, which is evidenced by PDOS. I think this is definitely not enough. The authors should at least study and compare the charging state and charge transfer of metal atoms on substrate. As there are many studies in using Cu atoms as SACs, what is the major advantage in using BP as substrate?

5) I have serious concerns over some computational methods, which may affect the results. The cut-off energy of 350 eV may be not enough. It is strange that the authors used a 4x4x2 k-point mesh for a slab system since a single point is enough for the Z direction. The authors should include spin-polarization for all calculations, as it will affect the energy and electronic structures.

On the experimental side,

1) The STEM images are too elusive to illustrate the single atom catalysts. On the reviewer's side, the BP flake is decorated with a lot of bright dots with different size. But it is hard to tell them are metal atoms.

2) The Raman spectra of as-prepared SACs on BP flakes should be provided.

3) The authors claimed that "The sample was calcined at 250°C in air for 15 min to remove organic ligands" (in SI Figure S8). This ultrastability is quite impressive because BP is quite sensitive to ambient conditions. The authors should comment the stability of BP after the incorporation of SACs.

COMMENTS TO AUTHOR:

Reviewer #1: The manuscript focuses on single atom catalyst synthesized from photochemical methods on a black phosphorous support, which results into high catalyst loading. They used different characterization methods including imaging techniques, high energy x-ray methods and inductively coupled plasma spectrometry to confirm the presence of the single atoms. There are a few concerns that need to be addressed before this work can be accepted.

1) On their synthesis scheme (Figure 3a), it is claimed that H₂ plays a role as both a hole scavenger to inhibit quick charge recombination and also as an oxidant to the precursor. Is this process only selective to H₂? Did the authors try other hole scavenging chemicals and also rule out the formation of M-H structures given they used these catalyst for HER?

Response: Thanks a lot for the reviewer's comments. In this work, we introduced hydrogen into the catalytic reaction system as a hole-trapping agent to inhibit the recombination of carriers, which effectively accelerated the kinetic process of metal ion reduction. Our choice of hole-trapping agent is mainly determined by the semiconductor properties of BP. In general, the electrode potential of the electron donor should be lower than the valence band values (E_{VB}) of BP (0.72 V). Some common hole trapping agents have been used in experiments, such as EDTA-2Na, trolamine (TEOA), or methanol. EDTA is excluded due to its easy complexation with metal ions. TEOA and methanol have both high redox potential, which are difficult to be directly oxidized by photogenerated holes, affecting photochemical reduction process. Importantly, the photoreduction process is carried out in organic solution and most inorganic hole trapping agents are not applicable in this system. H₂ hydrogen has strong reducibility and suitable redox potential, which meets the requirements of experimental conditions.

In addition, in the lab, it is hard to directly detect the existence of the M-H structures by some detection methods. Though we cannot completely rule out the formation of M-H structures, M-H structures are difficult to be stable in a reaction system. H[•] generated by photogenic holes has high activity as reaction intermediates and can be rapidly converted into H⁺ or H₂. Otherwise, the obtained single-atom catalysts will undergo a series of washing and drying treatments before electrochemical testing, so did not consider the effect of formation of M-H structures on HER process.

2) On Figure S11, the difference between single atom group and metal cluster should be correlated with EXAFS, where in the metal cluster case, the clear M-M bond is expected.

Response: We really appreciate your valuable opinions for us and have added relevant data according to the reviewer's advice. **Fig. R1a** shows the Cu K edge XANES spectra of Cu foil and Cu/BP with different Cu loading. The near-edge features of Cu/BP are in between of those of Cu foil and Cu₂O, indicating that the Cu species are partially positively charged ($Cu^{\delta+}$, $0 < \delta < 1$) due to the charge redistribution between Cu⁰ and BP. Fourier-transformed k^2 -weighted EXAFS in R space shows that Cu_{3.93}/BP and Cu_{11.3}/BP possess one main peak at 1.92 Å from the first coordination shell of Cu-P bond (**Fig R1b**). Cu_{15.8}/BP displays an additional minor peak at 2.4 Å, ascribed to Cu-Cu scattering, confirming the formation of Cu clusters. By contrast, no obvious peaks at 2.4 Å for Cu_{11.3}/BP and Cu_{3.93}/BP evidence that Cu atoms are atomically dispersed, in accordance with the HAADF-STEM observations (**Supplementary Fig. 12**). We have added the **Fig R1** in **Supplementary Fig. 13**, and the relevant discussion has been also included.

Fig. R1 a Cu k-edge XANES. b FT k^2 -weighted EXAFS spectra of Cu/BP and the reference Cu foil and Cu_2O .

3) From their EXAFs analysis Co oxidation state is +2, does this imply their photochemical method does not reduce Co as compared to Cu?

Response: Thank you for pointing this out, and we shall be pleased to elaborate on your questions. XANES of Co K edge in n-Co/BP reveals a high average valence close to +2 in **Fig. 2d**. Such a different oxidation state between Cu and Co is ascribed to a higher reducibility of Co atoms. By comparing these two FT-EXAFS of Co in n-Co/BP and Co^{2+}/BP (**Fig. R2**), we can see that the EXAFS spectrum of R space for n-Co/BP exhibits a dominant Co-P coordination at 1.71 \AA , unlike that of Co^{2+}/BP , which has multiple coordination shells at 1.60, 2.60, and 3.68 \AA , respectively. Thus, the coordination environment of Co single atom on BP surface is obviously different from that of Co^{2+} adsorbed by electrostatic force. In addition, the corresponding energy level diagrams of BP and Cu/BP are shown in **Supplementary Fig. 22**. It can be seen that the reduction potential of Cu/Co metal ions is higher than the conduction band edge of BP, so the photocatalytic reduction process can proceed in thermodynamics. **Fig. R3** compares the LSV activity of Co^{2+}/BP and n-Co/BP catalysts. It can be seen that Co^{2+}/BP has very poor catalytic activity, indicating that the high activity of n-Co/BP originates from the active sites of Co-P coordination structure. So Co^{2+} can be successfully reduced by photochemical method.

Fig. R2 a The normalized Co K-edge XANES and b FT-EXAFS spectra of n-Co/BP, Co foil and CoO. c The normalized Co K-edge XANES and d FT-EXAFS spectra of Co²⁺/BP, Co foil and CoO.

Fig. R3 HER performances of Co²⁺/BP and n-Co/BP.

4) What was the rationale chosen to select a cutoff energy of 350eV and the k-point densities for optimization and electronic properties? Were these values properly converged, if so, to what tolerance.

Response: The rationale chosen to select a cutoff energy of 350eV and the k-point densities for optimization and electronic properties is to establish the minimization of the electronic energy and the stabilization of the geometric structure of the system. Inspired by the previous work, ENCUT specifies the cutoff energy for the plane-wave-basis set in eV, and controls different number of plane waves at each k-point. The default value of cutoff energy is equal to largest ENMAX on the POTCAR file^[1]. The value of k point is based on an empirical formula, $k \propto a^{-3}$ for simple metals, $k \propto a^{-3}$ for semiconductors, $k \propto a^{-3}$ for insulators, where k represents the number of k points, a represents lattice constant^[2]. These values are properly converged and will run up to the set cutoff energy and k-point densities.

References

- [1] A H. J. Monkhorst, J. D. Pack, *Phys Rev B* **1976**, 13, 5188-5192; b P. Wisesa, K. A. McGill, T. Mueller, *Phys Rev B* **2016**, 93.

- [2] W. S. Morgan, J. E. Christensen, P. K. Hamilton, J. J. Jorgensen, B. J. Campbell, G. L. W. Hart, R. W. Forcade, *Computational Materials Science* **2020**, 173, 109340.

Reviewer #2: In this work, authors have reported Cu P3 single-atom catalysts non-strongly coordinated on BP support at mild temperatures. This material was prepared at a high loading of more than 11%, of Cu loading. The prepared catalyst is well characterized by several techniques including TEM, STEM, HAADF-STEM, FT-EXAFS and XANES, and other important characterization. Also, DFT studies have been explored in detail to prove the experimental investigations.

Some important comments:

1. The term “high electronegative ligand” is not clear at all. It is not clear, why the author calls it “high electronegative L “with respective what? What is the logic behind this design?

Response: We really appreciate your valuable opinions for us. The ligand represents the atoms, molecules and ions that can bond with the central atom (metal or metal-like). Compounds formed by central atoms and ligands become complexes. Here, we realize that the word “ligand” is improperly used, and there is ambiguity in its expression, so it is changed into “atoms”. The logic behind the design is described as follow.

Single-atom catalysts (SACs) with 100% metal dispersity offer the maximum atom efficiency, providing the most ideal strategy to create cost-effective catalysts. In general, the aggregation of SACs is typically attributed to the migration of atomic metal species due to weak metal–support interactions or strong metal–metal interaction.^[1] Introducing suitable ligands to the surface of the support to stabilize SACs is an effective approach to enhance the metal–support interactions, which can inhibit the migration tendency of metal SAs.^[2]

Various ligand strategies have been proposed to enhance the metal–support interactions. However, some common ligand atoms, such as N, O, and S, have high electronegativity, which easily leads to the coordinated SACs that are highly oxidized.^[3] For some catalytic reactions the electronic state of the catalytic centers probably determines the reactant adsorption, activation, formation of intermediate, and product desorption.^[4] Especially for catalytic reduction reactions, an electron-rich center is beneficial to the reduction of reactant.^[4c, 5] However, the high-electronegativity atoms probably poison or deactivate the SAC centers, which thus limits their application in catalysis. Hence, some low-electronegativity atoms are potential ligand candidates for constructing the electron-rich SAC centers with optimized electronic properties and chemical activity. BP with relatively low electronegativity and buckled structure favors the strong confinement of metal single atoms, which is particularly attractive for catalysis.^[6]

2. What about the reusability of these catalysts? any structural changes post reusability by XANES and HAADF-STEM to prove that the nature of SACs is stable?

Response: We really appreciate rigorous thinking of the reviewer. The stability of n-Cu/BP was estimated by chronoamperometric test under constant current of 10 mA cm⁻² and 50 mA cm⁻² in **Fig. 4h**, showing a stable overpotential for 22 h. In contrast, When the electrode works at 10 mA cm⁻² (current provided from BP was subtracted) for only 16.6 hours, a rapid increase in overpotential of 74 mV was observed, which was mainly ascribed to the easily oxidized surface of BP by oxygen in water when no metal atoms was loaded in BP. Moreover, after 2500 cyclic-voltammetry (CV) cycles, the LSV curves and corresponding mass activities of n-Cu/BP at different overpotential were presented in **Supplementary Fig. 32 and 33**. n-Cu/BP still retains 90% of its original electrocatalytic activity, indicating its considerable electrochemical stability.

The HAADF-STEM images and element mapping of n-Cu/BP after long-term electrocatalysis are shown in **Fig. R4,5 (Supplementary Fig. 34 and 35)**. The dense single-atom group is still presented without any aggregations for n-Cu/BP catalyst. The XANES and FT-EXAFS spectra of n-Cu/BP after long-time operation shows that the single-atom Cu sites remain atomic dispersion without aggregation (**Fig R6**), further demonstrating the stability of n-Cu/BP. The relevant discussion has been included in **Supplementary Fig. 36**.

Fig. R4 The image of n-Cu/BP by aberration-corrected transmission electron microscope after long-term electrocatalysis.

Fig. R5 HAADF-STEM image and element mapping of n-Cu/BP after long-term electrocatalysis.

Fig. R6 XAS characterizations after long-time operation. a XANES and b corresponding FT-EXAFS spectra for n-Cu/BP post HER, Cu foil, Cu_2O , and CuO.

3. Though, the author mentioned that high TOF, how it can be compared to previous procedures? A separate comparison table should be included in the SI. There are several

easier procedures that are already reported with better results, as this preparation procedure is quite complex.

Response: The turnover frequency (TOF) quantifies the specific activity of a catalyst center for a special reaction under defined reaction conditions by the number of molecular reactions. In different catalytic models, due to different catalytic mechanisms per unit active site, there is a large gap in TOF values. In this work, TOF is mainly used to explain the synergistic catalysis between neighboring single atoms, although this result is not prominent compared with other previous articles. In detail, the TOF of n-Cu/BP at an overpotential of 150 mV was calculated to be $0.53 \text{ H}_2 \text{ s}^{-1}$, which was three times higher than that of bi-atomic CuCo/BP ($0.17 \text{ H}_2 \text{ s}^{-1}$) and four times than of n-Co/BP ($0.12 \text{ H}_2 \text{ s}^{-1}$) (**Fig.4c**). The result indicates that neighboring single-atom Cu sites possess better catalytic kinetics than neighboring Co or CuCo sites on BP surface. In addition, A positive correlation between neighboring Cu or Co single atom composition and TOF suggests that the activity of n-Cu or n-Co is much higher than that of isolated single atom (**Fig.4e**). A detailed comparison of TOF values (**Table R1, Supplementary Table 7**) shows the high activity of our n-Cu/BP.

Table R1. Comparison of TOF of HER catalysts at overpotential 0.15 V in alkaline condition.

Catalysts	TOF ($\text{H}_2 \text{ S}^{-1}$)	References
n-Cu/BP	0.57	This work
CuCo/BP	0.17	This work
n-Co/BP	0.12	This work
Co ₁ /PCN	0.22	Nat. Catal. 2, 134 (2018)
Ru@GnP	0.145 (0.1 V)	Adv. Mater. 30, 1803676 (2018)
Co-Ni ₃ N	0.146 (0.2 V)	Chem. Sci. 2018, 9, 1822
Ni-MoS ₂	0.32	Energy Environ. Sci. 9, 2789-2793 (2016)
MoNi ₄ /MoO _{3-x}	1.13 (0.1 V)	Adv. Mater. 29, 1703311 (2017)
Mo _{0.1} Ni ₁ C ₂	0.465	Angew. Chem. Int. Ed. 56, 16086 (2017)
Ru/NG	0.35 (0.1 V)	ACS Appl. Mater. Interfaces 9, 3785-3791 (2017)
Ni-C-N NSs	0.44 (0.1 V)	J. Am. Chem. Soc. 138, 14546-14549 (2016)
Co-NiS ₂	0.55 (0.1 V)	Angew. Chem. Int. Ed. 58, 18676 (2019)

4. Some previous literature on Cu/ Co SACs and others should be included as respective section, e.g Chemical Reviews, 2021, 121,13620-13697; Small, 2021, 2006477; Coordination Chemistry Reviews Volume 418, 2020, 213376; <https://doi.org/10.1093/nsr/nwy077>; Small, 2021, 2006477; Advanced Materials Interfaces, 2021, 2001822; <https://doi.org/10.1021/acsnm.1c02743>.

Response: We acknowledge the reviewer for the positive suggestion. These recent related articles have a good reference for our work, and we have cited the above references in the relevant part of the article.

5. what about the practicability of these catalysts?

Response: Single-atom catalysts (SACs) offer the maximum atom efficiency, providing the most ideal strategy to create cost-effective catalysts. During the past decade, various concepts for atomic dispersion of metals on solid supports have emerged, such as utilization of vacancy defects on supports^[3c, 7], fabrication of metal-organic frameworks (MOFs)^[3b, 8], spatial confinement in zeolites^[9], and enhancement of the metal-support interactions^[3e, 10]. However, these routes involve fussy synthetic steps and sensitive conditions, including adsorption of metal precursors, followed by reduction and stabilization on supports^[11]. Moreover, especially under high amounts of metal precursors or high-temperature pyrolysis, these methods still cannot strictly exclude metal aggregation, resulting in low reproducibility^[12]. Currently most of SACs have reported very low metal loading, so the development of a practical and direct approach for fabricating SACs with high metal loading is particularly attractive in the field^[13].

Therefore, we report here a room-temperature photochemical strategy with hydrogen auxiliary to produce the stable and high-loading SACs, which provides a simple and practical synthesis route for SACs. In addition, the reported non-noble catalyst with dense neighboring Cu single-atom structure shows a low overpotential of only 41 mV at 10 mA cm⁻² and Tafel slope of 53.4 mV dec⁻¹ under alkaline hydrogen evolution reaction (HER), significantly surpassing commercial Pt/C. After 2500 cyclic-voltammetry (CV) cycles, n-Cu/BP still retains 90% of its original electrocatalytic activity, indicating its considerable electrochemical stability. In summary, its simple and low-cost synthesis method as well excellent HER activity meets the requirements of practical application.

References

- [1] a C. Z. Zhu, S. F. Fu, Q. R. Shi, D. Du, Y. H. Lin, *Angew Chem Int Edit* **2017**, *56*, 13944-13960; b A. Aitbekova, L. H. Wu, C. J. Wrasman, A. Boubnov, A. S. Hoffman, E. D. Goodman, S. R. Bare, M. Cargnello, *J Am Chem Soc* **2018**, *140*, 13736-13745; c A. Q. Wang, J. Li, T. Zhang, *Nat Rev Chem* **2018**, *2*, 65-81.
- [2] a Y. Tang, Y. T. Li, V. Fung, D. E. Jiang, W. X. Huang, S. R. Zhang, Y. Iwasawa, T. Sakata, L. Nguyen, X. Y. Zhang, A. I. Frenkel, F. Tao, *Nat Commun* **2018**, *9*, 1231; b Y. R. Xue, B. L. Huang, Y. P. Yi, Y. Guo, Z. C. Zuo, Y. J. Li, Z. Y. Jia, H. B. Liu, Y. L. Li, *Nat Commun* **2018**, *9*, 1460.
- [3] a R. Jiang, L. Li, T. Sheng, G. F. Hu, Y. G. Chen, L. Y. Wang, *J Am Chem Soc* **2018**, *140*, 11594-11598; b J. Wang, Z. Q. Huang, W. Liu, C. R. Chang, H. L. Tang, Z. J. Li, W. X. Chen, C. J. Jia, T. Yao, S. Q. Wei, Y. Wu, Y. D. Lie, *J Am Chem Soc* **2017**, *139*, 17281-17284; c J. Zhang, X. Wu, W. C. Cheong, W. X. Chen, R. Lin, J. Li, L. R. Zheng, W. S. Yan, L. Gu, C. Chen, Q. Peng, D. S. Wang, Y. D. Li, *Nat Commun* **2018**, *9*, 1002; d H. L. Li, L. B. Wang, Y. Z. Dai, Z. T. Pu, Z. H. Lao, Y. W. Chen, M. L. Wang, X. S. Zheng, J. F. Zhu, W. H. Zhang, R. Si, C. Ma, J. Zeng, *Nat Nanotechnol* **2018**, *13*, 411-+; e G. Vile, D. Albani, M. Nachtegaal, Z. P. Chen, D. Dontsova, M. Antonietti, N. Lopez, J. Perez-Ramirez, *Angew Chem Int Edit* **2015**, *54*, 11265-11269.
- [4] a X. Wang, W. X. Chen, L. Zhang, T. Yao, W. Liu, Y. Lin, H. X. Ju, J. C. Dong, L. R. Zheng, W. S. Yan, X. S. Zheng, Z. J. Li, X. Q. Wang, J. Yang, D. S. He, Y. Wang, Z. X. Deng, Y. E. Wu, Y. D. Li, *J Am Chem Soc* **2017**, *139*, 9419-9422; b P. Zhou, F. Lv, N. Li, Y. L. Zhang, Z. J. Mu, Y. H. Tang, J. P. Lai, Y. G. Chao, M. C. Luo, F. Lin, J. H. Zhou, D. Su, S. J. Guo, *Nano Energy* **2019**, *56*, 127-137; c X. Zhou, Q. Shen, K. D. Yuan, W. S. Yang, Q. W. Chen, Z. H. Geng, J. L. Zhang, X. Shao, W. Chen, G. Q. Xu, X. M. Yang, K. Wu, *J Am Chem Soc* **2018**, *140*, 554-557.
- [5] S. Back, J. Lim, N. Y. Kim, Y. H. Kim, Y. Jung, *Chem Sci* **2017**, *8*, 1090-1096.
- [6] C. Chen, W. Ou, K. M. Yam, S. B. Xi, X. X. Zhao, S. Chen, J. Li, P. Lyu, L. Ma, Y. H. Du, W. Yu, H. Y. Fang, C. H. Yao, X. Hai, H. M. Xu, M. J. Koh, S. J. Pennycook, J. L. Lu, M. Lin, C. L. Su, C. Zhang, J. Lu, *Adv Mater* **2021**, 2008471.
- [7] a R. Lang, W. Xi, J. C. Liu, Y. T. Cui, T. B. Li, A. F. Lee, F. Chen, Y. Chen, L. Li, L. Li, J. Lin, S. Miao, X. Y. Liu, A. Q. Wang, X. D. Wang, J. Luo, B. T. Qiao, J. Li, T. Zhang, *Nat Commun* **2019**, *10*, 234; b J. W. Wan, W. X. Chen, C. Y. Jia, L. R. Zheng, J. C. Dong, X. S. Zheng, Y. Wang, W. S. Yan, C. Chen, Q. Peng, D. S. Wang, Y. D. Li, *Adv Mater* **2018**, *30*, 1705369.

- [8] X. X. Wang, D. A. Cullen, Y. T. Pan, S. Hwang, M. Y. Wang, Z. X. Feng, J. Y. Wang, M. H. Engelhard, H. G. Zhang, Y. H. He, Y. Y. Shao, D. Su, K. L. More, J. S. Spendelow, G. Wu, *Adv Mater* **2018**, 1706758.
- [9] a M. Yang, S. Li, Y. Wang, J. A. Herron, Y. Xu, L. F. Allard, S. Lee, J. Huang, M. Mavrikakis, M. Flytzani-Stephanopoulos, *Science* **2014**, 346, 1498-1501; b Q. M. Sun, N. Wang, T. J. Zhang, R. Bai, A. Mayoral, P. Zhang, Q. H. Zhang, O. Terasaki, J. H. Yu, *Angew Chem Int Edit* **2019**, 58, 18570-18576.
- [10] a J. Deng, H. B. Li, J. P. Xiao, Y. C. Tu, D. H. Deng, H. X. Yang, H. F. Tian, J. Q. Li, P. J. Ren, X. H. Bao, *Energ Environ Sci* **2015**, 8, 1594-1601; b L. Wang, M. X. Chen, Q. Q. Yan, S. L. Xu, S. Q. Chu, P. Chen, Y. Lin, H. W. Liang, *Sci Adv* **2019**, 5:eaax6322.
- [11] a Y. T. Qu, Z. J. Li, W. X. Chen, Y. Lin, T. W. Yuan, Z. K. Yang, C. M. Zhao, J. Wang, C. Zhao, X. Wang, F. Y. Zhou, Z. B. Zhuang, Y. Wu, Y. D. Li, *Nat Catal* **2018**, 1, 781-786; b J. Lin, A. Q. Wang, B. T. Qiao, X. Y. Liu, X. F. Yang, X. D. Wang, J. X. Liang, J. X. Li, J. Y. Liu, T. Zhang, *J Am Chem Soc* **2013**, 135, 15314-15317.
- [12] a Y. H. He, S. Hwang, D. A. Cullen, M. A. Uddin, L. Langhorst, B. Y. Li, S. Karakalos, A. J. Kropf, E. C. Wegener, J. Sokolowski, M. J. Chen, D. Myers, D. Su, K. L. More, G. F. Wang, S. Litster, G. Wu, *Energ Environ Sci* **2019**, 12, 250-260; b C. Chu, D. Huang, S. Gupta, S. Weon, J. Niu, E. Stavitski, C. Muhich, J. H. Kim, *Nat Commun* **2021**, 12, 5179.
- [13] Y. Zhou, X. Tao, G. Chen, R. Lu, D. Wang, M. X. Chen, E. Jin, J. Yang, H. W. Liang, Y. Zhao, X. Feng, A. Narita, K. Mullen, *Nat Commun* **2020**, 11, 5892.

Reviewer #3: Authors of this work present a combined experimental and computational study on the preparation and characterization of single-atom catalysts on exfoliated black phosphorous flakes with ultrahigh atomic loading. Hydrogen was used as a hole-trapping agent for the preparation of SACs, which can improve the atomic loading and inhibit the aggregation of metal atoms. The authors show that the as-prepared SACs exhibit excellent HER activities. Here, I will mainly comment on the computational part of the manuscript, for which I have the following major concerns,

Response: We have revised the computational part according to the reviewer's advice in the revised manuscript. We improved the accuracy of the calculation method and recalculated and analyzed the previous data. The kinetic energy cut-off was set 450 eV in all computations to describe all atoms' valence electrons, and the 4×4×1 k-points grid was employed for electronic property computations. The questions and suggestions of the author have been carefully modified to improve the professionalism and scientificity of the article.

1) First, it is not clear why the authors choose and compare elements of Cu, Co, Fe, Ni, Mo, and Pt, since there are so many metal atoms in the periodical table. It is not clear why the low binding energy of metal atom is related to the free energies of hydrogen adsorption. What is "obvious up-shifting Fermi level" (compared to what?) Meanwhile, it is not clear how this binding energy is defined. I would suggest "adsorption energy" from the calculated results, as all data are negative.

Response: Thank you for pointing this out, and we shall be pleased to elaborate on your questions.

1) We introduced transition metal atoms to decorate BP nanosheets, which provide extra electrons to activate the H₂ molecules. The transition metals have to be screened since only the SACs with well-balanced empty/occupied d orbitals can exhibit optimal catalytic performance. For workload reasons, we selected a small number of metal atoms (Cu, Co, Fe, Ni, Mo, and Pt) for screening mainly according to one of the following two reasons. i), these metal atoms and their derivatives are common HER-active materials, especially their phosphates, which often show excellent water decomposition activity and have been widely studied. ii), BP has the adjustable direct-band-gap properties, enabling to work as an efficient photocatalyst with broadband solar absorption. Under visible light irradiation, the charges are generated to drive the in-situ reduction of metal ion adsorbed on BP surface. According to the photochemical property of BP, in theory, we can achieve the photochemical reduction of all metal ions with the reduction potential above -0.4 V vs. RHE. The selected metal elements meet the requirements of catalyst synthesis in the experiment. Based on the above considerations, we selected these metal elements for screening research.

2) We compared the binding energy and Fermi level of these transition metal single-atom catalysts as the correlativity of the ΔG_{H^*} , trying to find some kind of internal connection between them. The results show that Cu and Co single atoms on BP support could be attributed to their common features that low binding energy of M-P bond and simultaneously higher Fermi level, which could be the reason for their optimal ΔG_{H^*} . The “obvious up-shifting Fermi level” compares other metal atoms, indicating that Cu and Co have higher Fermi levels, which is not properly expressed here. The “obvious up-shifting” should be changed to “higher”.

In addition, we also realized that associating ΔG_{H^*} with binding energy in Fig 5a was subjective and unscientific, so we showed these data separately for illustration. The Fermi energy levels was replaced by the work function as the abscissa in **Fig R7**. The work function is defined as follows: $\Phi = E_{vac} - E_F$, where E_F is the Fermi energy, and E_{vac} is the electrostatic potential of the vacuum level (**Table R2**).

The relevant expressions will be amended as follows. The ΔG_{H^*} at single-atom Cu/BP catalyst is 0.020 eV, which are closer to the optimal value (i.e., 0 eV) and superior to the benchmark value of Pt (i.e., -0.089 eV) and Co/BP (i.e., 0.105 eV). The single-atom absorption energies (ΔE) were calculated subsequently. Compared to other metal single atoms, the Cu atoms absorbed on BP surface possess higher ΔE value (-2.905 eV), indicating their relatively weak interactions. Moreover, the feature that the lowest work function for Cu/BP suggests its obvious higher Fermi level, which leads to lower occupation of the anti-bonding states between the active sites and H^* intermediates, and therefore a stronger adsorption.

3) The binding energy (E_b) between metal single atom and the supported substrate was defined as follows:

$$E_b = E_{total}(M/substrate) - E_{total}(M) - E_{total}(substrate)$$

Where, $E_{total}(M/substrate)$, $E_{total}(M)$, and $E_{total}(substrate)$ are the total energy of M/BP, the single atoms, and the BP substrate containing a boron vacancy, respectively.

The adsorption energy (E_{ads}) of metal single atoms was calculated as follows:

$$E_{abs} = E_{total}(M + substrate) - E_{total}(M) - E_{total}(substrate)$$

in which $E_{total}(M + substrate)$, $E_{total}(M)$, and $E_{total}(substrate)$ are the total energy of BP with adsorbed metal single atom, metal single atom, and BP substrate, respectively.

Therefore, here the calculation formula of “binding energy” and “adsorption energy” is the same, representing the same meaning. We will explain the calculation formula in the paper.

Fig. R7 The Gibbs free energy (ΔG_{H^*}) of hydrogen adsorption sites for Cu, Co, Fe, Ni, Mo, and Pt single atoms on BP support and those corresponding calculated work functions (Φ) (inset: single-atomic adsorption energy (ΔE) on BP surface).

Table R2. The values of Fermi level, vacuum level, and work function of metal single atoms on BP surface.

	E_F (eV)	E_{vac} (eV)	Φ (eV)
Pt	-1.486	3.378	4.864
Mo	-1.370	3.314	4.684
Cu	-0.785	3.340	4.125
Fe	-1.308	3.282	4.590
Co	-1.050	3.591	4.641
Ni	-1.402	3.309	4.711

2) The authors claim “non-strong coordination”, and there are quite large binding energies. Is this a contradictory? In literature, it was revealed that there are strong covalent bonding between Cu/Co and BP. Please refer to Adv. Mater. 2021, 33, 2008471. Both the metal atoms and BP flakes are stable after 2500 cycles of HER.

Response: Catalytic metal sites supported on oxides or carbonaceous materials are usually strongly coordinated by oxygen or heteroatoms. These high-electronegative heteroatoms (N, O, and S) prevent effectively metal atomic agglomeration to achieve atomic-level dispersed catalytic structures, while these strong coordination bonds naturally cause an electron-deficient state for coordinated metal atoms and consequently affect their catalytic activity^[1]. Therefore, it is a critical challenge to construct support materials that can stabilize catalytic metal atoms without the aid of strong heteroatom coordination. The low-electronegativity P atoms are potential ligand candidates for constructing the electron-rich SACs centers^[2].

The stability of the metal-support composite could be quantitatively measured by the binding energy of single atom on the support. The more negative value of binding energy indicates that the single metal atom is attached to the BP more stably. In **Fig. 5a**, one can see that the binding energy of Cu on BP is -2.305 eV, which is the highest among the six catalysts. The distance between metal atom and adjacent P atom is significantly correlated with the binding energy. Cu/BP is the one with the largest distance and weakest binding of metal atom, which could be attributed to the relatively stable $3d^{10}4s^1$ outermost shell. As seen in **Fig. R8**, the Fourier-transformed (FT) k^2 -weighted EXAFS of Cu in n-Cu/BP shows a main peak at around 1.92 Å, corresponding to the first coordination shell Cu–P coordination. By contrast, n-Co/BP shows a dominant peak around 1.71 Å, different from that of Co–O bond (1.60 Å) in CoO. The Cu–P bond is significantly larger than the Co–P bond. Moreover, the M–P bond is also larger than the M–O bond. To sum up, the Cu–P bond in n-Cu/BP possesses the non-strong coordination structure with a low interaction.

In addition, we have made reference to this paper (Adv. Mater. 2021, 33, 2008471). Similar to the work in the reference, the near-edge features of Cu/BP are in between of those of Cu foil and Cu_2O , indicating that the Cu species are partially positively charged ($Cu^{\delta+}$, $0 < \delta < 1$) due to the charge redistribution between Cu^0 and BP. The intrinsic coordination properties of P atoms with low electronegativity in BP are also used to make it an ideal platform to support low-valence single metal atoms without doping other heteroatoms. However, this reference is also obviously different from our work. “The few-layer BP flakes were heated up to 150 °C in atomic layer deposition (ALD) chamber to generate extra vacancies as the anchor sites. Pd atoms are anchored in the divacancy of BP via covalent bonding. Due to the existence of vacancies, Pd atom confined in the vacancies of BP to form four Pd–P bonds (V–Pd–P₄). In contrast, the XANES spectrum simulates the Pt atomic structures and shows two oxygen atoms into the local coordination structure to form P₂–Pt–O₂–P₂.” Therefore, both the presence of P-vacancy and oxygen will strengthen the binding energy of metal atoms on BP surface, forming the strong coordination structures.

In our work, although single-atom Cu anchored on BP surface has the strong coordination structure, the peculiar puckered structure of BP could provide a cage-like coordination environment to efficiently bond single atoms with M–P₃ coordination in the confined space. After 2500 cyclic-

voltammetry (CV) cycles, n-Cu/BP still retains 90% of its original electrocatalytic activity, indicating its considerable electrochemical stability.

Fig. R8 FT-EXAFS spectra of n-Cu/BP (a) and n-Co/BP (b).

References:

- [1] a S. Li, B. B. Chen, Y. Wang, M. Y. Ye, P. A. van Aken, C. Cheng, A. Thomas, *Nat Mater* **2021**, *20*, 1240. b P. Zhou, N. Li, Y. G. Chao, W. Y. Zhang, F. Lv, K. Wang, W. X. Yang, P. Gao, S. J. Guo, *Angew Chem Int Edit* **2019**, *58*, 14184-14188.
- [2] H. Y. Fang, C. H. Yao, X. Hai, H. M. Xu, M. J. Koh, S. J. Pennycook, J. L. Lu, M. Lin, C. L. Su, C. Zhang, J. Lu, *Adv Mater* **2021**, 2008471.

3) The authors should know the Fermi level is different for different systems. So have they considered the alignment of all energies to Vacuum level? If not, the Fermi level plots (Figs. 5a and 5b) will be meaningless.

Response: We really appreciate rigorous thinking of the reviewer. We have aligned them to the vacuum levels and converted them into work functions for analysis when considering Fermi levels. The Fermi energy levels was replaced by the work function as the abscissa in **Fig 5a**. The work function is defined as follows: $\Phi = E_{vac} - E_F$, where E_F is the Fermi energy, and E_{vac} is the electrostatic potential of the vacuum level (**Supplementary Table 9**).

4) The authors highlight that the metal atoms are electron rich, which is evidenced by PDOS. I think this is definitely not enough. The authors should at least study and compare the charging state and charge transfer of metal atoms on substrate. As there are many studies in using Cu atoms as SACs, what is the major advantage in using BP as substrate?

Response: Thanks a lot for the reviewer's comments. In this work, after coupling single-atom Cu with BP, the charge density in hybrid's interlayer is redistributed in the form of an apparent electron transfer from Cu atom to BP, and an electron-rich region around Cu atom is formed (**Fig. R9**). More importantly, the calculated position of density of states (PDOS) of these common metal monatomic catalysts on BP surfaces are shown in **Fig. 5b**. We can see that most 3d states of Cu/BP are localized below the Fermi level and are filled with electrons. In particular, the ε_d of only Cu single atom at BP surface locates at a more high-lying position than Pt, implying the electron-rich feature of the Cu/BP.

In addition, we used BP as substrate mainly for the following reasons. Despite the significant effort in avoiding agglomeration of single atoms while increasing single atomic loading, the traditional high amounts of metal precursors or high-temperature pyrolysis still cannot strictly exclude metal aggregation, which is difficult to meet the above two conditions. The high-electronegative heteroatoms (N, O, and S) prevent effectively metal atomic agglomeration to achieve atomic-level dispersed catalytic structures, while these strong coordination bonds naturally cause an electron-deficient state for coordinated metal atoms and consequently affect their catalytic activity. Therefore, the development of a practical and direct approach for fabricating SACs with high metal loading and electron-rich state is particularly attractive in the field.

Unlike other planar 2D materials such as graphene and $\text{-C}_3\text{N}_4$, and MoS_2 , the intrinsic coordination properties of phosphorus atoms with a relatively low electronegativity in BP potentially render them as an ideal platform to support low-valence single-metal atoms without additional heteroatom doping. Furthermore, BP has the adjustable direct-band-gap properties, enabling to work as an efficient photocatalyst with broadband solar absorption. Based on this feature, we report a room-temperature photochemical strategy with hydrogen auxiliary to produce the stable and high-loading SACs on BP support. By introducing H_2 as a hole-trapping agent into the in-situ photochemical reduction process, a significantly increased single-atom loading was revealed. In theory, with this method, we can achieve the synthesis of all metal atoms with a reduction potential above -0.4 V vs. RHE . In addition, the peculiar puckered structure of BP provides a cage-like coordination environment, which makes it possible to load high-loading metal single atoms.

Fig. R9 The differential charge density diagram of Cu single-atom supported on BP. Green and orange contours represent electron accumulation and depletion. The isosurface is $0.002\text{ e}/\text{\AA}^3$.

5) I have serious concerns over some computational methods, which may affect the results. The cut-off energy of 350 eV may be not enough. It is strange that the authors used a $4\times 4\times 2$ k-point mesh for a slab system since a single point is enough for the Z direction. The authors should include spin-polarization for all calculations, as it will affect the energy and electronic structures.

Response: We really appreciate rigorous thinking of the reviewer. We improved the accuracy of the calculation method and recalculated and analyzed the previous data. With fixed cell parameters, the model structures were fully optimized using the convergence criteria of 10^{-5} eV for the electronic energy and $0.03\text{ eV}/\text{\AA}$ for the forces on each atom and the plane wave cutoff was set to 450 eV, and the $4\times 4\times 1$ k-points grid was employed for electronic property computations. In addition, the spin-polarization has been considered in previous calculations and will be specified in the calculation methods. As seen in **Fig. R10**, the relevant data have been revised to be more rigorous and are discussed in this paper. Since there is no obvious difference between the two results, we still retain the conclusion obtained from the previous calculation.

Fig. R10 Theoretical calculations of n-Cu/BP on HER. a The Gibbs free energy (ΔG_{H^*}) of hydrogen adsorption sites for Cu, Co, Fe, Ni, Mo, and Pt single atoms on BP support and those corresponding calculated work functions (Φ) (inset: single-atomic adsorption energy (ΔE) on BP surface). b The projected d band density of states (PDOS) for common transition metal monatomic model on BP surfaces. The position of d band center is indicated by the short black horizontal bar. The gray solid line indicates the Fermi level. The gray dash line indicates the d band center of Pt. c Schematic illustration of water dissociation in alkaline solutions for n-Cu/BP catalyst. d Barrier energy of water dissociation diagram for n-Cu/BP, n-Co/BP and CuCo/BP.

On the experimental side,

1) The STEM images are too elusive to illustrate the single atom catalysts. On the reviewer's side, the BP flack is decorated with a lot of bright dots with different size. But it is hard to tell them are metal atoms.

Response: Thanks a lot for the reviewer's comments, and we understand your concern. There are two main reasons for blurring STEM images. i) BP has poor electron beam tolerance. In the HAADF mode, the structure of BP is easily destroyed by the electron beam. Therefore, these STEM data were not carefully tuned in the shooting process, resulting in unclear images. ii) There is little difference in relative atomic mass between P and Cu or Co atoms, which makes it difficult to distinguish metal atoms from BP substrates in STEM images. Therefore, we re-tested the samples and added the clearer STEM images to better distinguish metal single atoms and improve the quality of the paper in **Fig. R11 (Supplementary Fig. 9)**.

Fig. R11 HAADF-STEM image of n-Cu/BP (a) and n-Co/BP (b).

2) The Raman spectra of as-prepared SACs on BP flacks should be provided.

Response: We really appreciate your valuable opinions for us. Raman spectroscopy can provide fingerprint information of substances, and shows great advantages in the study at solid–liquid interfaces and the detection of species at low-wavenumber regions, such as oxygen species and hydroxy. However, Raman can only monitor the structure information of the bulk catalyst owing to the low sensitivity^[3]. In addition, Raman is not sensitive to the detection of metal and single metal atoms, impeding the application of Raman spectroscopy on SACs. Surface-enhanced Raman spectroscopy (SERS) possesses extremely high sensitivity and can provide a variety of structural information of trace species on the catalyst surface^[4]. Bifunctional nanostructures integrating catalysts and plasmonic substrates are usually required to enable SERS enhancement on the catalyst surface. However, this method requires high instrument sensitivity, and the mechanism of spectral signal enhancement is still being explored, which makes this method not widely used. Since the SERS test conditions cannot be obtained, and the plasmonic substrates matching with the n-Cu/BP catalyst has not been explored, it is difficult to provide the Raman spectra of as-prepared metal single atoms. In the lab, we can only display the fingerprint information of BP substance in n-Cu/BP catalyst by Raman spectra. Nevertheless, we used different characterization methods including STEM, FT-EXAFS and XANEs, and other important characterization to confirm the presence of the single atoms with M-P3 structure on BP support.

References

- [3] a H. K. Lee, C. S. Koh, W. S. Lo, Y. J. Liu, I. Y. Phang, H. Y. Sim, Y. H. Lee, C. P. Q. Gia, X. M. Han, C. K. Tsung, X. Y. Ling, *J Am Chem Soc* **2020**, *142*, 11521-11527; b H. Y. F. Sim, H. K. Lee, X. Han, C. S. L. Koh, G. C. Phan-Quang, C. L. Lay, Y. C. Kao, I. Y. Phang, E. K. L. Yeow, X. Y. Ling, *Angew Chem Int Edit* **2018**, *57*, 17058-17062.
- [4] a T. Hartman, R. G. Geitenbeek, G. T. Whiting, B. M. Weckhuysen, *Nat Catal* **2019**, *2*, 986-996; b K. F. Zhang, L. Yang, Y. F. Hu, C. H. Fan, Y. R. Zhao, L. Bai, Y. L. Li, F. X. Shi, J. Liu, W. Xie, *Angew Chem Int Edit* **2020**, *59*, 18003-18009; c J. Wei, S. N. Qin, J. Yang, H. L. Ya, W. H. Huang, H. Zhang, B. J. Hwang, Z. Q. Tian, J. F. Li, *Angew Chem Int Ed* **2021**, *60*, 9306-9310.

3) The authors claimed that “The sample was calcined at 250°C in air for 15 min to remove organic ligands” (in SI Figure S8). This ultrastability is quite impressive because BP is quite sensitive to ambient conditions. The authors should comment the stability of BP after the incorporation of SACs.

Response: Typically, the samples go through a necessary “organic impurity removal” step before testing STEM to prevent contamination of the electron probe. It is usually carried out in two ways: low temperature calcination or electron beam cleaning. Our samples were calcined at 250°C to remove impurities before STEM testing. According to the literatures, pure BP is prone to oxidation under the condition of coexistence of water and oxygen, while BP loaded with metal ions or single atoms has stronger stability because metal atoms replace the oxygen adsorption sites and strengthen the BP structure^[5]. In spite of this, the sample still retains good crystal structure and atom-dispersed single atom, as seen in **Supplementary Fig. 8**. In order to avoid readers’ misunderstanding, we

modify this sentence to “the samples go through a “organic impurity removal” step before testing STEM to obtain stable signals”.

In addition, the stability of n-Cu/BP was estimated by chronoamperometric test under constant current of 10 mA cm^{-2} and 50 mA cm^{-2} in **Fig. 4h**, showing a stable overpotential for 22 h. In contrast, When the electrode works at 10 mA cm^{-2} (current provided from BP was subtracted) for only 16.6 hours, a rapid increase in overpotential of 74 mV was observed, which was mainly ascribed to the easily oxidized surface of BP by oxygen in water when no metal atoms was loaded in BP. Moreover, after 2500 cyclic-voltammetry (CV) cycles, the LSV curves and corresponding mass activities of n-Cu/BP at different overpotential were presented in **Supplementary Fig. 32 and 33**. n-Cu/BP still retains 90% of its original electrocatalytic activity, indicating its considerable electrochemical stability.

The HAADF-STEM images and element mapping of n-Cu/BP after long-term electrocatalysis are shown in **Supplementary Fig. 34 and 35**. The dense single-atom group is still presented without any aggregations for n-Cu/BP catalyst. The XANES and FT-EXAFS spectra of n-Cu/BP after long-time operation shows that the single-atom Cu sites remain atomic dispersion without aggregation (**Fig R12**), further demonstrating the stability of n-Cu/BP. The relevant discussion has been included in **Supplementary Fig. 36**. The Raman shifts of n-Cu/BP and BP nanosheets after 2500 CV cycles are collected in **Supplementary Fig. 37**. n-Cu/BP still maintains these three typical peaks corresponding to Raman spectra of BP. The pure BP is transformed into red phosphorus or phosphorus oxide (PO_x) after long-time electrocatalysis. These results suggest that the BP substrate loaded with metal single atom has strong oxidation resistance and stable structure.

Fig. R12 XAS characterizations after long-time operation. a XANES and b corresponding FT-EXAFS spectra for n-Cu/BP post HER, Cu foil, Cu_2O , and CuO.

References

- [5] a Z. Hu, Q. Li, B. Lei, Q. Zhou, D. Xiang, Z. Lyu, F. Hu, J. Wang, Y. Ren, R. Guo, E. Goki, L. Wang, C. Han, J. Wang, W. Chen, *Angew Chem Int Ed Engl* **2017**, *56*, 9131-9135; b Z. Guo, S. Chen, Z. Wang, Z. Yang, F. Liu, Y. Xu, J. Wang, Y. Yi, H. Zhang, L. Liao, P. K. Chu, X. F. Yu, *Adv Mater* **2017**, *29*, 1703811; c D. Liu, J. Wang, J. Lu, C. Ma, H. Huang, Z. Wang, L. Wu, Q. Liu, S. Jin, P. K. Chu, X. F. Yu, *Small Methods* **2019**, *3*, 1900083.

REVIEWER COMMENTS

Reviewer #1 (Remarks to the Author):

The authors have addressed my concerns on the manuscript and thus I recommend publication of this work in Nature Communications.

Reviewer #2 (Remarks to the Author):

Authors have addressed all my comments and this revised version of the manuscript can be acceptable in Nature Communications.

Reviewer #3 (Remarks to the Author):

I have carefully read the response and revised version of the manuscript, and found that the authors cannot properly treat the issues that I raised, especially the theoretical part. Meanwhile, I have serious concerns over the validity of the computational results, as I show in the following,

- 1) First and foremost, both Reviewer #1 and Reviewer #3 have raised concerns over some key computational settings, such as cutoff energy, k-point sampling and spin polarization. However, the authors never show the convergence of these settings to obtain reliable results. The authors stated that they have used spin-polarized calculations, but the data shown in Fig. 5 are all nonmagnetic (even the DOS), which makes the results not trustable. As I mentioned before, spin polarization will affect the energy and electronic structures. I hope the authors know what they are doing on this point.
- 2) For "binding energy" and "adsorption energy", it is generally accepted that the two terms have opposite signs. I do not know why the author argue that "calculation formula of "binding energy" and "adsorption energy" is the same". This is rather confusing.
- 3) I believe the authors are not clear on the calculated results of the Fermi level and work function. "the feature that the lowest work function for Cu/BP suggests its obvious higher Fermi level" is not reasonable, because work function comes from Fermi level according to the definition.
- 4) The authors tried to use charge density redistribution to see the transfer, which is qualitative but not enough. If the authors plotted the charge density redistribution for other systems, they may get the same results. Meanwhile, if "the charge density in hybrid's interlayer is redistributed in the form of an apparent electron transfer from Cu atom to BP", then why "an electron-rich region around Cu atom is formed"? This is quite contradictory.
- 5) The authors emphasize that the proposed structure is featured with non-strong coordination, but they also say that "In our work, although single-atom Cu anchored on BP surface has the strong coordination structure". This is also contradictory. I turn to believe that the term "non-strong coordination" is not valid for the proposed system.

Based on the above facts, I think the paper cannot be accepted. The experiment may be valuable, but the paper contains basic flaws from a science point of view.

COMMENTS TO AUTHOR:

Reviewer #1 and Reviewer #2: Authors have addressed all my comments and this revised version of the manuscript can be acceptable in Nature Communications.

Response: We are very grateful to the reviewers for their recognition of the article. our comments make the article more rigorous and scientific.

Reviewer #3: I have carefully read the response and revised version of the manuscript, and found that the authors cannot properly treat the issues that I raised, especially the theoretical part. Meanwhile, I have serious concerns over the validity of the computational results, as I show in the following,

1) First and foremost, both Reviewer #1 and Reviewer #3 have raised concerns over some key computational settings, such as cutoff energy, k-point sampling and spin polarization. However, the authors never show the convergence of these settings to obtain reliable results. The authors stated that they have used spin-polarized calculations, but the data shown in Fig. 5 are all nonmagnetic (even the DOS), which makes the results not trustable. As I mentioned before, spin polarization will affect the energy and electronic structures. I hope the authors know what they are doing on this point.

Response: We really appreciate your valuable opinions for us. The important modifications have been executed to the theoretical calculation part of the paper in order to better verify and explain our views. The convergence of these settings was shown to obtain reliable results. As shown in **Fig. R1**, when ENCUT is greater than 450 eV and k-point exceeds $2 \times 2 \times 1$, the energy fluctuations are lower, so we adopt a cutoff energy of 450eV and a $2 \times 2 \times 1$ grid centered at the gamma (Γ) point for all calculations.

Fig. R1 Using different K points (a) and different encut (b) to calculate the average atomic energy of BP.

Furthermore, via careful investigation, we have revised the DOS. As shown in **Fig. R2**, the redrawn DOS contains both spin-up and spin-down. The calculated magnetic moment values are listed as below:

	Co/BP	Cu/BP	Fe/BP	Mo/BP	Ni/BP	Pt/BP
mag	1.0000	0.0001	2.0004	0.0000	0.0002	0.0000

Fig. R2 Total and partial DOS of Cu, Co, Fe, Ni, and Mo single atoms on BP support.

2) For “binding energy” and “adsorption energy”, it is generally accepted that the two terms have opposite signs. I do not know why the author argue that “calculation formula of “binding energy” and “adsorption energy” is the same”. This is rather confusing.

Response: Based on the reviewer’s suggestion, we changed “binding energy” to “adsorption energy” in the article as seen in **Fig 5b**. The adsorption energy (E_{abs}) between metal single atom and the supported substrate was defined as follows:

$$E_{abs} = E_{total}(M + substrate) - E_{total}(M) - E_{total}(substrate)$$

in which $E_{total}(M + substrate)$, $E_{total}(M)$, and $E_{total}(substrate)$ are the total energy of BP with adsorbed metal single atom, metal single atom, and BP substrate, respectively.

3) I believe the authors are not clear on the calculated results of the Fermi level and work function. “the feature that the lowest work function for Cu/BP suggests its obvious higher Fermi level” is not reasonable, because work function comes from Fermi level according to the definition.

Response: Thanks a lot for the reviewer’s comments. In this work, we removed the statement of the calculated results of the Fermi level and work function because it was inappropriate and redundant. Therefore, we added the total and partial DOS calculations of Cu, Co, Fe, Ni, and Mo single atoms on BP support. As shown in **Fig. 5a**, Cu/BP exhibits the lowest conduction band minimum (CEM), indicating its strongest reduction ability. The Fermi level crosses the conduction band of Cu/BP. This phenomenon indicates higher the electron mobility in Cu/BP structure, which has an important effect on the electrocatalytic HER¹⁻².

4) The authors tried to use charge density redistribution to see the transfer, which is qualitative but not enough. If the authors plotted the charge density redistribution for other systems, they

may get the same results. Meanwhile, if “the charge density in hybrid’s interlayer is redistributed in the form of an apparent electron transfer from Cu atom to BP”, then why “an electron-rich region around Cu atom is formed”? This is quite contradictory.

Response: Thank you for pointing this out, and we shall be pleased to elaborate on your questions. **Fig. 5a** displays the transition metals (Mo, Cu, Fe, Co, and Ni) on BP support of total and 3d/4d orbitals density of state (DOS). Cu/BP exhibits the lowest conduction band minimum (CEM), indicating its strongest reduction ability. The Cu 3d state is located in a more negative region with the most negative d-band center (-3.515 eV) than those in other transition metals on BP, suggesting that the Cu sites show the more enriched electron state on Cu/BP³⁻⁴. The charge density analysis in **Fig. 5c** shows that the incorporation of heteroatoms has appreciable influence on electron distribution. After coupling single atoms with BP, the charge density in hybrid’s interlayer is redistributed in the form of an apparent electron transfer from those metal atoms to BP. By compared from recently published studies on electron-defect/-rich single-atom Cu catalysts with different coordination structures in **Supplementary Table R1**, the largest bond length of Cu-P and lowest Bader charge value for n-Cu/BP catalyst demonstrates its weak non-strong interaction between Cu and BP as well as electron-rich properties of Cu site.

Supplementary Table R1. Comparison of bond lengths and Bader charges of single-atom Cu-based catalysts.

Catalysts	Coordination	Bond length by FT-EXAFS (Å)	Bader charge (e ⁻)	Reference
n-Cu/BP	Cu-P ₃	1.93	-0.30	This work
Cu-N ₄ /C	Cu-N ₄	1.42	-0.62	5
Cu-N ₄ /C-B		1.44	-0.89	
Cu-N ₄ /C-P		1.47	-0.59	
Cu ₂ @C ₃ N ₄	N-Cu-N	1.62	-0.66	6
(Zn, Cu)-NC	Cu-C ₂ N	1.71(Cu-N)	-0.60	4
	Zn-N ₄	1.44	-1.16	
Cu-SA/SNC	Cu-N ₄	1.44	-1.14	7
PdCu/NC	Cu-N ₂	1.47	-0.56	8
Cu-CDs	Cu-N ₂	1.50	-0.87	9
	Cu-O ₂	1.50		
CuNi-DSA/CNFs	CuN ₄	1.50	-0.69	10
	NiN ₄	1.52	-0.66	

5) The authors emphasize that the proposed structure is featured with non-strong coordination, but they also say that “In our work, although single-atom Cu anchored on BP surface has the strong coordination structure”. This is also contradictory. I turn to believe that the term “non-strong coordination” is not valid for the proposed system.

Response: The stability of the metal-support composite could be quantitatively measured by the binding energy of single atom on the support. The more negative value of binding energy indicates that the single metal atom is attached to the BP more stably. In **Fig. 5b**, one can see that the binding energy of Cu on BP is -2.305 eV, which is the highest among the five catalysts. The distance between metal atom and adjacent P atom is significantly correlated with the binding energy. Cu/BP is the one with the largest distance and weakest binding of metal atom, which could be attributed to the relatively stable 3d¹⁰4s¹ outermost shell. As seen in **Fig. R3**, the Fourier-transformed (FT) *k*²-weighted EXAFS of Cu in n-Cu/BP shows a main peak at around 1.92 Å, corresponding to the first coordination shell Cu-P coordination. By contrast, n-Co/BP shows a dominant peak around 1.71 Å, different from that of Co-O bond (1.60 Å) in CoO. The Cu-P bond is significantly larger than the Co-P bond. Moreover, the M-P bond is also larger than the M-O bond. Therefore, the Cu-P bond in n-Cu/BP possesses the non-strong coordination structure with a low interaction.

Fig. R3 FT-EXAFS spectra of n-Cu/BP (a) and n-Co/BP (b).

Reference

- Xu, W.; Chen, C.; Tang, C.; Li, Y.; Xu, L., Design of Boron Doped C₂N-C₃N Coplanar Conjugated Heterostructure for Efficient HER Electrocatalysis. *Sci Rep* **2018**, *8* (1), 5661.
- Liu, Y.; Wu, J.; Hackenberg, K. P.; Zhang, J.; Wang, Y. M.; Yang, Y.; Keyshar, K.; Gu, J.; Ogitsu, T.; Vajtai, R.; Lou, J.; Ajayan, P. M.; Wood, Brandon C.; Yakobson, B. I., Self-optimizing, highly surface-active layered metal dichalcogenide catalysts for hydrogen evolution. *Nature Energy* **2017**, *2* (9), 17127.
- Zhou, P.; Zhang, Q. H.; Xu, Z. K.; Shang, Q. Y.; Wang, L.; Chao, Y. G.; Li, Y. J.; Chen, H.; Lv, F.; Zhang, Q.; Gu, L.; Guo, S. J., Atomically Dispersed Co-P-3 on CdS Nanorods with Electron-Rich Feature Boosts Photocatalysis. *Adv. Mater.* **2020**, *32* (7), 1904249.
- Deng, D. J.; Qian, J. C.; Liu, X. Z.; Li, H. P.; Su, D.; Li, H. N.; Li, H. M.; Xu, L., Non-Covalent Interaction of Atomically Dispersed Cu and Zn Pair Sites for Efficient Oxygen Reduction Reaction. *Adv. Funct. Mater.* **2022**, 2203471.
- Zhou, X.; Ke, M. K.; Huang, G. X.; Chen, C.; Chen, W. X.; Liang, K.; Qu, Y. T.; Yang, J.; Wang, Y.; Li, F. T.; Yu, H. Q.; Wu, Y. E., Identification of Fenton-like active Cu sites by heteroatom modulation of electronic density. *P Natl Acad Sci USA* **2022**, *119* (8).
- Xie, P. F.; Ding, J.; Yao, Z. H.; Pu, T. C.; Zhang, P.; Huang, Z. N.; Wang, C. H.; Zhang, J. L.; Zecher-Freeman, N.; Zong, H.; Yuan, D. S.; Deng, S. W.; Shahbazian-Yassar, R.; Wang, C., Oxo dicopper anchored on carbon nitride for selective oxidation of methane. *Nat Commun* **2022**, *13* (1), 1375.
- Jiang, Z. L.; Sun, W. M.; Shang, H. S.; Chen, W. X.; Sun, T. T.; Li, H. J.; Dong, J. C.; Zhou, J.; Li, Z.; Wang, Y.; Cao, R.; Sarangi, R.; Yang, Z. K.; Wang, D. S.; Zhang, J. T.; Li, Y. D., Atomic interface effect of a single atom copper catalyst for enhanced oxygen reduction reactions. *Energ Environ Sci* **2019**, *12* (12), 3508-3514.
- Han, L. L.; Ren, Z. H.; Ou, P. F.; Cheng, H.; Rui, N.; Lin, L. L.; Liu, X. J.; Zhuo, L. C.; Song, J.; Sun, J. Q.; Luo, J.; Xin, H. L. L., Modulating Single-Atom Palladium Sites with Copper for Enhanced Ambient Ammonia Electrosynthesis. *Angew Chem Int Edit* **2021**, *60* (1), 345-350.
- Cai, Y. M.; Fu, J. J.; Zhou, Y.; Chang, Y. C.; Min, Q. H.; Zhu, J. J.; Lin, Y. H.; Zhu, W. L., Insights on forming N,O-coordinated Cu single-atom catalysts for electrochemical reduction CO₂ to methane. *Nat Commun* **2021**, *12* (1), 586.
- Hao, J.; Zhuang, Z.; Hao, J.; Wang, C.; Lu, S.; Duan, F.; Xu, F.; Du, M.; Zhu, H., Interatomic Electronegativity Offset Dictates Selectivity When Catalyzing the CO₂Reduction Reaction. *Adv Energy Mater* **2022**, 2200579.

REVIEWER COMMENTS

Reviewer #3 (Remarks to the Author):

In the response and revised version of the manuscript, I see that the authors have made efforts in revising the theoretical part. Most of the issues that I raised have been addressed, but there are still places that are scientifically incorrect, or expressions/statements that are not rigorous. Here, I would like to help the authors clarify some of these points.

- 1) From Fig. R1 in the Response, the ENCUT energy is OK, but the kpoint of $2 \times 2 \times 1$ seems not converged. I suggest the authors check the adsorption energy of metal atoms, free energy of hydrogen adsorption and the energy barrier with respect to denser kpoints. My experience is that for a 3×3 supercell of phosphorene, a kpoint of $2 \times 2 \times 1$ is not enough. Also, I do not think the authors can use data like "-3.515", "-2.305", "0.634", et.c, as the significant digit lies within the error caused by the parameters used for calculation (such as kpoints).
- 2) On page 6, I suggest use "indicating their relatively weak interactions compared to other atoms". Normally, an adsorption energy of ~ -2.3 eV is not weak but strong chemical bonding. Depending the system, you may call it weak or relative weak with adsorption energies, say, $-0.2 \sim -0.8$ eV.
- 3) One page 7, "Single-atom Cu site has a moderate M-H interaction and its Bader charge is only $-0.3 e^-$." This is scientifically wrong. The Bader charge of Cu should not be $-0.3 e^-$. From my understanding, $-0.3 e^-$ is the charge transfer.
- 4) The statement, "Cu/BP exhibits the lowest conduction band minimum (CEM), indicating its strongest reduction ability", is not scientifically rigorous. You may say lowest conduction band or conduction band minimum, but nobody understands the lowest conduction band minimum. Also, from the DOS, the Fermi level crosses conduction band, making the system metallic. Moreover, conduction band minimum is CEM or CBM?
- 5) In Fig. 5(a), there are some places that the peaks of total DOS are smaller than that of d-projected DOS, especially near Fermi level. This is not reasonable, and should be revised. Meanwhile, it is better to clarify that "M-t" stands for total and "M-d" for d states. Without this, "M-t" is confusing. In Fig. 5(e), revise to "Cu loses $-0.3 e^-$ "?

I thus strongly suggest the authors ask experts in theoretical computation to double check the presentation on the theoretical part (probably in their future work). This may help avoid scientific errors and misleading points that confuse readers.

COMMENTS TO AUTHOR:

Reviewer #3: In the response and revised version of the manuscript, I see that the authors have made efforts in revising the theoretical part. Most of the issues that I raised have been addressed, but there are still places that are scientifically incorrect, or expressions/statements that are not rigorous. Here, I would like to help the authors clarify some of these points.

1) From Fig. R1 in the Response, the ENCUT energy is OK, but the kpoint of $2\times 2\times 1$ seems not converged. I suggest the authors check the adsorption energy of metal atoms, free energy of hydrogen adsorption and the energy barrier with respect to denser kpoints. My experience is that for a 3×3 supercell of phosphorene, a kpoint of $2\times 2\times 1$ is not enough. Also, I do not think the authors can use data like “-3.515”, “-2.305”, “0.634”, et.c, as the significant digit lies within the error caused by the parameters used for calculation (such as k points).

Response: Thank you for pointing this out, a new k-points test is done with cutoff energy set to 450 eV. As shown in Fig. R1, when the k-points exceed $3\times 3\times 1$, the energy fluctuation per P atom is small within 1 mV/atom, so we recalculated the adsorption energy, density of states, and transition states, etc. using $3\times 3\times 1$ k-points to get the more accurate results. According the calculated results, we revised them in the Fig. 5 and Supporting Information.

Fig. R1 The k-points with the energy per atom of BP.

2) On page 6, I suggest use “indicating their relatively weak interactions compared to other atoms”. Normally, an adsorption energy of ~ -2.3 eV is not weak but strong chemical bonding. Depending the system, you may call it weak or relative weak with adsorption energies, say, -0.2 \sim -0.8 eV.

Response: Thanks for your suggestion. The manuscript is corrected with your suggested statement. Although the Cu atom absorbed on BP surface possesses strong bonding with a high ΔE value (-2.43 eV), it is relatively weak interactions compared with other candidates.

3) One page 7, “Single-atom Cu site has a moderate M-H interaction and its Bader charge is only -0.3 e-.” This is scientifically wrong. The Bader charge of Cu should not be -0.3 e-. From my understanding, -0.3 e- is the charge transfer.

Response: We really appreciate your valuable opinions for us. According to our recalculation results, the Bader charge analysis shows a positive charge of 0.29 on Cu. It means 0.29 |e| charge transfer occurs from the Cu atom to the BP layer.

4) The statement, “Cu/BP exhibits the lowest conduction band minimum (CEM), indicating its strongest reduction ability”, is not scientifically rigorous. You may say lowest conduction band or conduction band minimum, but nobody understands the lowest conduction band

minimum. Also, from the DOS, the Fermi level crosses conduction band, making the system metallic. Moreover, conduction band minimum is CEM or CBM?

Response: Thanks a lot for the reviewer's comments. Indeed, the fermi level crosses the DOS through our study means that the catalyst shows the metallic property. For electrocatalytic hydrogen evolution reaction, superior electrical conductivity facilitates reaction efficiency. Furthermore, we delete the improper statement for CBM in the manuscript.

5) In Fig. 5(a), there are some places that the peaks of total DOS are smaller than that of d-projected DOS, especially near Fermi level. This is not reasonable, and should be revised. Meanwhile, it is better to clarify that "M-t" stands for total and "M-d" for d states. Without this, "M-t" is confusing. In Fig. 5(e), revise to "Cu loses -0.3 e-"

Response: Thank you for pointing this out, we revised the DOS and charge density differences figure based on your suggestion. And legend of Total and d-project DOS is corrected with "Total" and "*d*". In the main text "Cu loses 0.3 e-" is corrected to "0.29 |e| charge transfer from Cu to substrate". And the arrow is used to show the electron transfer is from the Cu atom to the BP layer.